

# Three main stages in the uplift of the Tibetan Plateau during the Cenozoic period and its possible effects on Asian aridification: A review

Zhixiang Wang[1,2*]    Yongjin Shen[1,2]   Zhibin Pang[3]

[1]*State Key Laboratory of Biogeology and Environmental Geology, School of Earth Sciences, China University of Geosciences, Wuhan 430074, China.*

[2]*Laboratory of Critical Zone Evolution, School of Earth Sciences, China University of Geosciences, Wuhan 430074, China*

[3]*Shanxi Geological Survey Institute*

*Corresponding authors:wangzhi8905@126.com

**Abstract:** The Tibetan Plateau uplift and its linkages with the evolution of the Asian climate during the Cenozoic are a research focus for numerous geologists. Here, a comprehensive review of tectonic activities across the Tibet shows that the development of the Tibetan Plateau has undergone mainly three stages of the uplift: the near-modern elevation of the central Tibet and significant uplift of the northern margins (~55-35 Ma), the further uplift of the plateau margins (30-20 Ma), and a rapid uplift of the plateau margins again (15-8 Ma). The first uplift of the plateau during ~55-35 Ma forced the long-term westward retreat of the Paratethys Sea. The high elevation of the central Tibet and/or the Himalayan would enhance rock weathering and erosion contributing to lowering of atmospheric $CO_2$ content, resulting in global cooling. The global cooling, sea retreat coupled with the topographic barrier effect of the Tibetan Plateau could have caused the initial aridification in central Asia during the Eocene time. The second uplift of the northern Tibet could have resulted in the onset of the East Asian winter monsoon as well as intensive desertification of inland Asia, whereas the central-eastern in China became wet. The further strengthening of the East Asian winter monsoon and the inland Asian aridification during 15-8 Ma was probably associated with the Tibetan Plateau uplift and global cooling. Therefore, the uplift of the Tibetan Plateau plays a very important role in the Asian aridification.

**Keywords:** Tibetan Plateau,Asian aridification, uplift ,India-Asia collision



## 1. Introduction

The collision between India and Asia during the Cenozoic period created the high Himalaya Mountains and the Tibetan Plateau, which profoundly affected the global Cenozoic climate (e.g., Garzione, 2008; Molnar et al., 2010; Raymo and Ruddiman, 1992) and the geochemical composition of the ocean as a result of input fluxes of dissolved salts from the Tibetan Plateau to the sea (e.g., Chatterjee et al., 2013; Misra and Froelich, 2012). Reconstructing the uplifting processes of the Tibetan Plateau and its relationship with crustal deformation is of wide-ranging importance to understand the lithospheric evolution, surface uplift and global climate changes.

The Tibetan Plateau covers an area of more than 2.5 million km$^2$ with an average elevation of about 5000 m. In general, the Tibetan Plateau consists of six nearly west-east stretching tectonic blocks including the Himalayan, Lhasa, Qiangtang, Songpan-Ganzi-Hoh-Xil, Kunlun-Qaidam and Qilian blocks from south to north, separated by Indus-Yarlung suture (IYS), Bangong-Nujiang suture (BNS), Jinshajiang suture (JS), Anyimaqen-Kunlun-Mutztagh suture (AKMS) and South Qilian suture (SQS), respectively (Li et al., 2015; Yin and Harrison, 2000) (Fig. 1). The Lhasa and Qiangtang blocks are characterized by the main flat plateau with an average elevation of about 5000 m including several sedimentary basins. The plateau margins are a series of orogenic belts, with an average elevation ranging from 5500 to 6500 m, and sedimentary basins, such as the Qilian Mountains and Qaidam basin to the north, the Longmen Shan and Sichuan basin to the east (Fig.1). Based on seismic velocity models and wide-angle seismic profiles, the average crust thickness was interpreted as about 70-75 km under the southern Tibet, ~60-65 km under the plateau margins, and approximately 36 to 40 km beneath the Sichuan basin to the east and Tarim basin to the north (Jiang et al., 2006; Owens and Zandt, 1997; Tseng et al., 2009; Wang et al., 2007).

The timing of the initial contact and main India-Asia collision is still ambiguous with suggestions ranging from 70 to 34 Ma (Aitchison et al., 2007; DeCelles et al., 2014; Ding et al., 2005; Hu et al., 2015; Leech et al., 2005; Meng et al., 2012; Najman et al., 2010; Van Hinsbergen et al., 2012; Zhu et al., 2013), and it is probable that the main collision was not simultaneous along the entire convergent belt. Van Hinsbergen et al. (2012) proposed a two-stage India-Asia collision with phases at ~52 and 25-20 Ma based on the compilation of palaeomagnetic data from Lhasa and Tethyan Himalaya terranes. Based on the radiolarian and nannofossil biostratigraphy coupled with detrital zircon U-Pb geochronology from the Sangdanlin region in south Tibet, Hu et al. (2015) suggested that the onset of the India-Asia collision was at 59±1 Ma. Provenance analysis from upper Cretaceous-Paleocene strata in the Tethys Himalaya was proposed for the closure time of the Neo-Tethys Ocean and the India-Asia collision between 70 and 58±0.6 Ma (Cai et al., 2011; DeCelles et al., 2014). However, most of evidence based on geological, geophysical and geochemical data indicates that the main Indian subcontinent-Asia collision occurred between 55 and 50 Ma inferred by the following reasons: 1) the initiation of substantial faunal exchange of medium-to large-sized mammals during 53.3-50 Ma or a little earlier between India and Asia



block (e.g., Clementz et al., 2010; Clyde et al., 2003); 2) the plate motion of India decreased
dramatically during 55-50 Ma (e.g., Guillot et al., 2003; Shellnutt et al., 2014; van Hinsbergen et
al., 2011b), indicating the initial India-Asia collision (Li et al., 2015) or the slab breakoff of the
subducting Neo-Tethyan oceanic lithosphere (Ji et al., 2016; Zhu et al., 2015); 3) a pronounced
flare up in magmatic activities and the ultrahigh-pressure metamorphism around 55-50 Ma (Ding
et al., 2016; Donaldson et al., 2013; Guan et al., 2012; Zhu et al., 2015); 4) the provenance change
of the Himalayan foreland basin resulted from the first arrival of the Lhasa detritus (Green et al.,
2008; Najman et al., 2010; Wang et al., 2011; Zhu et al., 2005); 5) Paleomagnetic studies show
that the initial contact of the India-Asia collision occurred around 55-50 Ma or a little earlier (e.g.,
Chen et al., 2010; Huang et al., 2015; Meng et al., 2012; Najman et al., 2010; Sun et al., 2010a).
Tan et al. (2010) proposed a younger collision age of 43 Ma based on the paleomagnetic results
from the late Cretaceous red beds, lava flows and Eocene tuffs in the Lhasa block, but Najman et
al. (2010) suggested that the sampled volcanic tuffs were a short-term large eruption that only
recorded a snapshot record of the Earth magnetic field at high inclination, therefore its
paleomagnetic inclination should be taken caution. Therefore, we preferred an age of ~55-50 Ma
of initial India-Asia collision in this study.

After the initiation of the India-Asia collision, the Tibetan Plateau has experienced two

basically deformational styles. One is N-S crustal shortening and the tectonic uplift of the
adjoining mountains. The E-W extension and related N-S trending rifts are another deformation
pattern of the plateau. These two deformational styles accommodated most of India-Asia
convergence. However, it is still uncertain how much of the total convergence between India and
stable Asia after their initial collision was absorbed by the crustal shortening and E-W extension
since the India-Asia collision (Dupont-Nivet et al., 2010; Guillot et al., 2003; Li et al., 2015; Tan
et al., 2010; van Hinsbergen et al., 2011a, 2011b; Yin and Harrison, 2000). Based on the available
paleomagnetic data, Guillot et al. (2003) estimated a total India-Asia convergence of 3215±496
km and ~1100 km shortening of Himalayan since 55 Ma. Dupont-Nivet et al. (2010) estimated
2900±600 km subsequent latitudinal convergence between India and Asia, divided into 1100±500
km within Asia and 1800±700 km within India inferred from the apparent polar wander paths of
India and Asia. Some paleomagnetic results indicate that the Himalayan region experienced at
least 1500±480 km of post-collisional crustal shortening and 2000±550 km within Asia since the
collision (Sun et al., 2010a). According to the marine magnetic anomalies and the Eurasia-India
plate circuit, van Hinsbergen et al. (2011a) argued that the convergence was up to 3200-4000 km
for the India-Asia collision since 55 Ma. Recently, Li et al. (2015) concluded that ~1630 km of
shortening occurred across the Tibetan Plateau with more than ~1400 km accommodated by large-
scale thrust belts since 55 Ma based on a comprehensive review of published geological and
simulated data. Although the amount of India-Asia convergence accommodated by the large-scale
thrust belts is still uncertain, the large-scale thrust belts not only contribute to the crustal
shortening in central Tibet but also cause the uplift of the plateau margins (e.g., DeCelles et al.,



2002; Li et al., 2015; Tapponnier et al., 2001; Yin and Harrison, 2000).
The continued uplift of the Tibetan Plateau profoundly influenced Cenozoic global and Asian
climate. Uplift of the Tibetan Plateau could have resulted in high rainfall on the front slopes of
Himalayas as a result of the more intense monsoonal circulation and the orographic barrier (e.g.,
Thiede et al., 2004). The high Tibetan-Himalaya orogen would lead to greater rates of silicate
weathering and erosion contributing to lowering of atmospheric $CO_2$ concentrations to force
global cooling (e.g., Dupont-Nivet et al., 2008a; Garzione, 2008; Raymo and Ruddiman, 1992).
Additionally, the high Tibetan Plateau and/or only the Himalayan mountains provided the
dominant heat source for the South Asian summer monsoon or orographic insulation, driving the
large-scale monsoon flow and simultaneously acting as an obstacle to southward flow of cool, dry
air (Boos and Kuang, 2010; Molnar et al., 2010; Wu et al., 2012). The rising Tibetan Plateau
disrupted global circulation of the westerly winds, shifting the smooth flow to the diverted flow
around the high plateau (e.g., Chatterjee et al., 2013). Numerous studies show that the uplift of the
Himalayan-Tibetan orogen is closely related to the onset of Asian monsoon system and Asian
desertification (e.g., Chatterjee et al., 2013; Guo et al., 2002; Miao et al., 2012; Zhang et al., 2007).
In this paper, we synthesize the available data to propose that there are three significant stages in
the uplift of the plateau and its possible effects on climatic changes in Asia.

## 2. Three main phases of growth of the Tibetan Plateau and Asian drying changes during the Cenozoic

Available deformational and paleoaltimetry data indicate that there were three main phases of
growth of the Tibetan Plateau since the India-Asia collision. These episodes caused regionally
climatic changes as well as contributing to trends in Cenozoic global cooling. The spatial and
temporal evolution of the plateau growth and effects on Asian climate are divided into three
episodes: the Eocene (~55-35 Ma), the middle Oligocene-early Miocene (30-20 Ma) and the
middle to late Miocene (15-8 Ma).

### 2.1. The significant uplift of the northern margins accompanied by Asian aridification between ~55 Ma and 35 Ma

#### 2.1.1. Asian initial aridification during ~55-35 Ma

Recent simulations show that although the high elevation of the central Tibet has already
been removed, the large-scale South Asian summer monsoon circulation was unaffected by
providing the high but narrow orography of the Himalaya and adjacent mountains (Boos and
Kuang, 2010). These mountains produced a strong monsoon by insulating warm, moist air over
continental India from the cold and dry extratropics (Boos and Kuang, 2010). Using an
atmospheric general circulation model with 1.9º longitude resolution with prescribed sea surface
temperature and sea ice cover to examine the effects of the plateau uplift on climate, the results
were in general agreement with Boos and Kuang (2010), suggesting that the uplift of the Himalaya

Ⓒ Author(s) 2018. CC BY 4.0 License.





would strengthen summer precipitation in southwestern margin of the Himalaya as well as central-
southern India (Zhang et al., 2012). The low oxygen isotope values with strong seasonality in
gastropod shells and mammal teeth from Myanmar at 40-34 Ma, and aeolian dust deposition in
northwest China during the Eocene time in response to the onset of desertification and winter
monsoon circulation in inner Asia show marked monsoon-like patterns in rainfall and wind south
and north of Tibetan-Himalayan orogen during the late Eocene time (Licht et al., 2014) and that
support the view that the Asian monsoon was probably active during the Eocene (Quan et al.,
2012). The similar fossil leaf trait spectra between Eocene basins in southern China and modern
Indonesia-Australia Monsoon suggest that the characteristics of the modern topographically
enhanced South Asia Monsoon had to develop in Eocene time (Spicer et al., 2016).
Sedimentological and numerical data shows that monsoons were not dampened by the Proto-
Paratethys Sea (Bougeois et al., 2018). The strong Eocene monsoons later weakened after 34 Ma
ago related to the global shift to icehouse climate (Licht et al., 2014).
The near-modern elevation of the central Tibet and further extension to the north probably
forced the long-term westward sea retreat from the Tarim Basin (e.g., Bosboom et al., 2014a;
Carrapa et al., 2015; Sun et al., 2016). The lithostratigraphic, biostratigraphic and
magnetostratigraphic results from the southwest Tarim Basin along the Pamir and West Kunlun
range show that the final sea retreat was between 47 and 40 Ma accompanied by significant
aridification of the Asian interior as a result of the decrease of moisture supplied from the
Paratethys Sea (Bosboom et al., 2014a, 2014b; Sun et al., 2016). Sedimentology, paleontology,
sandstone petrography and zircon U-Pb ages from the Tajik depression, 400 km to the west of the
Tarim basin, show that the local retreat of this part of the Paratethys Sea was at ~39 Ma, a little
later than the Tarim Basin (Carrapa et al., 2015). A strong anticyclonic zone at Central Asian
latitudes and an orographic effect from emerging Tibetan Plateau occurred during this period
(Bougeois et al., 2018). These results are in agreement with the northward growth of the Pamir
Mountains.
In the Xining basin at the northeastern of Tibetan Plateau, the palynological records show a
sudden appearance of the Pinaceae family at 38 Ma in response to the cooler and drier climate,
and suggest that the initiation of the continental aridification in central Asia started as early as
Eocene time (Dupont-Nivet et al., 2008a). Subsequent studies of the same sedimentary sequence
in Xining basin reveal second additional phases of aridification before the Eocene-Oligocene
Transition (34 Ma). The first phase at~36.6 Ma was accompanied by a distinct decrease in gypsum
content relative to red mudstone and the second phase was characterized by a substantial increase
in clastic sedimentation rate at 34.7 Ma (Abels et al., 2011). At the Eocene-Oligocene Transition,
playa lake deposits in Xining basin vanished, subsequent dominated by homogenous red
mudstones with minor interstitial gypsum content, in response to a pronounced aridification of the
Xining basin (Dupont-Nivet et al., 2007).



### 2.1.2. Tectonic uplift of the Tibetan Plateau linked to this aridification

Previous studies indicate that the Lhasa and Qiangtang terranes underwent significant crustal thickening and surface uplift prior to the India-Asia collision (DeCelles et al., 2002; Li et al., 2015). Shortening reconstructions estimate that a ~60% crustal shortening of the Lhasa block occurred during the Cretaceous and gained 3-4 km of elevation prior to the India-Asia collision (Murphy et al., 1997). Balanced cross section restoration across the Qiangtang block suggests that ~400 km of crustal shortening occurred prior to the India-Asia collision (Li et al., 2015; van Hinsbergen et al., 2011b). The majority of intensive shortening across Central Tibet occurred before the collision based on the structural restorations, and this region has been affected by only minimal thrusting reactivation since the late Paleocene (Kapp et al., 2003, 2005). Therefore, Central Tibet (Lhasa and Qiangtang terranes) attained at least 3 km elevation prior to India-Asia collision. Since the India-Asia collision, the significant crustal thickening (~160 km) of the central Tibet only occurred within about 10 Myr. The northward subduction of Greater India slab played a major role in crustal thickening and uplifting (Li et al., 2015). The southward subduction of the Songpan-Ganzi terrane beneath the Qiangtang block also contributed to the crustal thickening of the central Tibet, as inferred from the widespread potassium-rich lavas in the northern Qiangtang (Ding et al., 2007; Li et al., 2015).

In the Qiangtang block, stable isotope results from fluvial/lacustrine carbonate cement, pedogenic carbonate and marl from the Kangtuo and Suonahu formations indicate that high elevation (> 5000 m) had been established by at least the middle Oligocene (28 Ma) (Fig 2; Xu et al., 2013). Stable isotopes revealed a paleoelevation of ~4.1-6.5 for the southern Tibet and 3.3 km for the southeast Tibet in the Eocene, respectively (Hoke et al., 2014; Ingalls et al., 2017). The low-temperature thermochronlogic results from the Qiangtang and Lhasa terranes showed a rapid to moderate exhumation between 85 and 45 Ma followed by low exhumation rates of <0.05 mm/yr, which explained the plateau formation in central Tibet by 45 Ma (Rohrmann et al., 2011). In addition, the distributions of high-K calc-alkaline andesites, dacites and rhyolites in central-western Qiangtang from 46 to 38 Ma, together with the north-south trending dikes in response to the onset of east-west extension in central Tibet between 47 and 38 Ma, suggested that the central Tibet had already attained near-modern elevation by at least 38 Ma (Wang et al., 2008, 2010). Thus, the Lhasa and Qiangtang terranes have reached near-modern elevation by at least 35 Ma.

The northern Tibetan Plateau had experienced significant uplift and exhumation between 55 and 35 Ma (Fig. 2). Low-temperature thermochronologic data shows that rocks along the major thrusts-the West Qinling thrust (Clark et al., 2010; Duvall et al., 2011), Qilian Shan (He et al., 2017), Tanggula thrust (Li et al., 2012), Fenghuoshan fold-thrust belt (Staisch et al., 2016), Kunlun fault (Jolivet et al., 2001), Altyn Tagh thrust (Jolivet et al., 2001; Yin et al., 2002) and Kashgar-Yecheng thrust (Cao et al., 2013) had undergone rapid cooling and exhumation between 55 and 40 Ma as a response to the initiation of India-Asia collision (Fig. 2; Locations have shown on the circles and detail information can be seen at table 1). Seismic reflection profiles and



balanced cross section restoration show that the compression of the northern Qaidam basin began
at 65-50 Ma, which was consistent with high accumulation rates of the foreland basin (Ji et al.,
2017; Wei et al., 2013; Yin et al., 2008a) (Fig. 2).
The strong uplift of the mountains in the plateau margins during this interval would offer a
large amount for clastic sediments to the adjacent basins, with peaks of influxes into the Lanzhou
basin at ~58 Ma (Wang et al., 2016b), Xining basin at ~52 Ma (Dai et al., 2006), and Hoh Xil
basin at ~52 Ma (Zhang et al., 2010). In the Tethyan Himalaya, emplacement of a series of
undeformed granitoid bodies before $44.1 \pm 1.2$ Ma indicates that significant crustal thickening had
occurred within 10 to 20 Myr of the initial India-Asia collision (Aikman et al., 2008). The low-
temperature thermochronologic data from the Deosai plateau in the northwest Himalaya coupled
with the thermal history modeling shows that the Deosai plateau underwent continuous slow
exhumation rates for the past 35 Ma, thus suggesting that the high elevation had been achieved by
at least 35 Ma (Fig 2; van der Beek et al., 2009). Therefore, the plateau margins have undergone
significant growth shortly after the initiation of India-Asia collision, but the altitude is still
disputed.
Although the Eocene global cooling that would reduce the amount of water vapor held in the
atmosphere was revealed by deep-sea stable oxygen isotope (Zachos et al., 2001), we consider that
the Tibetan Plateau uplift at this period played an important role in Asian aridification. First,
climate models suggest that surface uplifts of the northern Tibetan Plateau had a greater
contribution to the decreased annual precipitation over inland Asia mainly due to the enhanced
rain shadow effect of the mountains and changes in the regional circulations (Liu et al., 2015a;
Zhang et al., 2017). Second, the outward-growth of the Tibetan Plateau would force westward sea
retreat of Paratethys Sea, resulting in decrease of moisture supplied into inland Asia.

### 2.2 The further uplift of the plateau margins and strengthened aridification in Asia between 30 and 20 Ma

#### 2.2.1. East Asian monsoon and strengthened aridification during 30-20 Ma

The Oligocene-Miocene transition is a significant Cenozoic cooling event referred to Mi-1
with a series of paleoenvironmental changes. Benthic foraminiferal oxygen isotope from the ODP
site 1218 in Pacific shows a transient ~1‰ positive excursion as a response to the expansion of
Antarctic ice sheets (Zachos et al., 2001; Pälike et al., 2006), and an apparent positive excursion of
benthic foraminiferal carbon isotope (Pälike et al., 2006) (Fig 3C and 3D). Sea level estimates
from coastal plain coreholes in New Jersey and Delaware show an about 50 m fall of sea level
between 22.3 and 23.3 Ma (Kominz et al., 2008) (Fig 3E). The $CaCO_3$ contents and the proportion
of > 150 μm (wt%) from ODP site 1264 and 1265 in the subtropical southeastern Atlantic Ocean
show significant increases between 22.2 and 23.2 Ma as a feedback to the transient Oligocene-
Miocene transition glaciations (Liebrand et al., 2016) (Fig 3F and 3G). The benthic foraminiferal
accumulation rates (BFAR) at the southern Atlantic site 1090 significantly increased during





Oligocene-Miocene transition period, imply an enhanced paleoproductivity (Diester-Haass et al.,
2011) (Fig 3H). Benthic foraminiferal Mg/Ca, Li/Ca and U/Ca records from ODP 926 and 929 in
the equatorial Atlantic across the Oligocene-Miocene boundary reveal an enhanced organic carbon
burial (Mawbey and Lear, 2013). However, the driving mechanism of this fundamental transition
is still ambiguous. The relatively stable atmospheric $CO_2$ content may not be the reason for this
climatic change (Fig 3B), and a minimum in eccentricity that results in low seasonality orbits
favorable to ice-sheet expansion on Antarctica may be a dominant factor (Zachos et al., 2001) (Fig
3A).
Significantly, there were also obvious changes in Asian paleoenvironments during this
interval. Based on a compilation of paleobotanical and lithological data from 125 cites over China,
Sun and Wang (2005) argued that a reorganization of climate system, from latitudinal zonal
pattern during the Paleogene to a Neogene pattern with arid zones restricted to northwest China,
occurred around the Oligocene-Miocene boundary. This implies that the onset of the East Asia
summer monsoon began around ~23 Ma. The continuous aeolian deposits during 22 to 6.2 Ma in
Qin'an county (Gansu province) support the conclusion that modern East Asian monsoon already
existed in the early Miocene (Fig 4; Guo et al., 2002). Subsequent studies from the Zhuanglang
site at the western Chinese Loess Plateau confirmed that the loess deposits in the Loess Plateau
began as early as 25 Ma and inland Asian desertification initiated or enhanced at least by the late
Oligocene (Fig 4; Qiang et al., 2011). A 30 Ma stable isotope record of marine-deposited black
carbon from the northern South China Sea reveals that $C_4$ plants gradually appeared since the
early Miocene as a component of land vegetation in East Asia; and this shift in vegetation types
might be related to the evolution of East Asian monsoon (Jia et al., 2003). The sporomorphs
results from the Lanzhou basin during the latest Early Oligocene indicate a dominance of arboreal
plants that represent a wetter environment characterized by relatively high precipitation and a
warm climate, which suggests that East Asia summer monsoon has already supplied abundant
rainfall to Lanzhou basin (Miao et al., 2013). Monsoonal circulation existed by the early Miocene
was also supported by the presence of persistently lower pedogenic carbonate $\delta^{13}C$ and higher soil
respiration fluxes on the Loess Plateau and in the Himalayan foreland (Caves et al., 2016).
Weathering records from the ODP 1148 in South China Sea and ODP 718 in Bay of Bengal reveal
an increased intensity of chemical weathering related to onset of East Asian summer monsoon
(Clift et al., 2008, 2014). The intensification of the South Asian monsoon at ~24 Ma was probably
a major trigger of the stronger erosion on Greater Himalayan with removal of ~1.5 km rocks
leading to a major unconformity in the Himalayan foreland basin (Clift and VanLaningham, 2010).
The aridification of Asian interior further intensified during the late Oligocene-early Miocene.
In Jungger basin, the earliest eolian deposition started at 24 Ma and lasted until 8 Ma, indicating
that extensive arid to semiarid regions existed in the Asian interior by 24 Ma (Sun et al., 2010b).
According to the radioisotopic methods ($^{40}Ar$-$^{39}Ar$ and U-Pb ages) to precisely date a volcanic tuff
preserved in the stratigraphy from the Aertashi and Kekeya sections in the Tarim basin, in



combination with the magnetostratigraphy and lithostratigraphy, Zheng et al. (2015) concluded
that the initial desertification of the Taklimakan desert was between ~26.7 Ma and 22.6 Ma as a
response to a combination of widespread regional aridification and increased erosion in the
surrounding mountain fronts, both of which were closely linked to the tectonic uplift of the
Tibetan-Pamir Plateau and Tian Shan. A palynological record from the fluviolacustrine Jingou
River section collected from the northern Tian Shan indicates a shift from a late Oligocene wet
condition in central Asia to dry conditions at 23.8-23.3 Ma (Tang et al., 2011). A significant
increase in aeolian sediments in Lanzhou basin occurred at ~26 Ma, which reveals that a large
scale arid environment formed in the Asian interior since the late Oligocene (Zhang et al., 2014).
In central Tibet, stable isotope analyses of modern and accurately dated ancient paleosol carbonate
in the Nima basin reveal an arid climate and high paleoelevation (4.5-5 km) by 26 Ma (DeCelles
et al., 2007). Major and trace element concentrations from the central Pacific show that the
delivery of Asian dust materials significantly increased since 20 Ma in the ODP Site 1215 (Ziegler
et al., 2007), which was compatible with the remarkable aridification of inland Asia.

**2.2.2. Tectonic uplifts of the Tibet and surrounding mountains linked to this drying**
This stage is characterized by relatively little tectonic active in the central Tibet and by
further uplift of the plateau margins (Fig. 4; Locations that mentioned the uplift and deformation
at this part have been shown on the circles and detail information can be seen table 2).
In northeastern Tibet, low-temperature thermochronologic results show that the Laji Shan
(Lease et al., 2011), Ela Shan (Lu et al., 2012) and northeastern Qilian (Pan et al., 2013)
underwent significant rapid cooling and exhumation between 25 and 20 Ma (Fig. 4). The unstable
accumulations in the Xining basin during 25-20 Ma (Xiao et al., 2012), high accumulation rates in
the Xunhua basin around 24-21 Ma (Lease et al., 2012) and sedimentary discontinuity in the
Guide basin at~21 Ma (Liu et al., 2013) have been interpreted to reflect the uplift of adjacent
mountains during this period. Changes in paleocurrent and detrital zircon provenance at ~30 Ma in
the Lanzhou basin at the northeast margin of the Tibetan Plateau reflect the pulsed growth of the
West Qinling (Wang et al., 2016b). In the northwest Tibet, the initiation of thrusting in the West
Kunlun Range began in the early Miocene (~23 Ma) (Jiang et al., 2013). The apatite fission track
results indicate that the Altyn Tagh fault (Jolivet et al., 2001), the Main Pamir thrust (Sobel and
Dumitru, 1997), the Southwest Tian Shan (Sobel et al., 2006), and the Northern Tian Shan
(Hendrix et al., 1994) underwent rapid cooling and exhumation between 30 and 20 Ma. All of
these indicate the initial activity of the thrust faults and a significant tectonic deformation of the
Tibet margins during the middle Oligocene-early Miocene time (Fig. 4).
In the Himalayas, low-temperature thermochronologic results in combination with the
leucogranite U-Pb and K-Ar muscovite ages show the formation of the Silving Rift as early as 23-
21 Ma (Searle et al., 1999). The initial thrusting of the Main Central Thrusts occurred at
approximately 23-21 Ma based on the geochronology from the dating of $^{40}Ar/^{39}Ar$ from the



Greater Himalayan paragneiss in hanging wall of the Main Central thrust (Robinson et al., 2006)
and was synchronous with the South Tibetan detachment system motion (Li et al., 2015; Robinson
et al., 2006). In eastern Tibet, low-temperature thermochronologic data reveals that the Longmen
Shan underwent significantly cooling during 30-25 Ma (Wang et al., 2012b). Therefore, we can
conclude that the plateau margins experienced intense growth between 30 and 20 Ma (Fig. 4).
The Paratethys Sea has retreated since late Eocene (Bosboom et al., 2014a), which is not the
main cause of this Asian aridification. Global cooling trends and changes in $CO_2$ level are unlikely
to account for this strengthened aridification because late Oligocene warming, as documented by
the marine $\delta^{18}O$ records (Zachos et al., 2001), are not correlative with this drying changes in Asia.
Therefore, we consider that the surface uplifts of the plateau margins are the dominant factor. The
continuing uplift and expansion of the plateau margins would alter significantly the thermally
forced circulation and enhance continental-scale winter monsoon and central Asian aridity (An et
al., 2001). Climate models reveal that uplift of the northern Tibet margins have significant effects
on the intensified drought in inland Asia (Liu et al., 2015a; Zhang et al., 2012). Another important
factor is the Tian Shan Mountains and surrounding mountains uplift, which would reduce westerly
moisture transport (Bougeois et al., 2018) and thus strengthen drying in central Asia.

### 2.3 The rapid uplift and erosion of the plateau margins again and Asian aridification between 15 and 8 Ma



#### 2.3.1. Strengthened Asian winter monsoon and extensive aridification during 15-8 Ma


The middle-late Miocene time was a fundamental change in earth's climate system. A
significant ~1‰ positive excursion of benthic foraminiferal $\delta^{18}O$ reflected a major expansion and
permanent establishment of the East Antarctic ice sheets, and an apparent positive excursion of
benthic foraminiferal $\delta^{13}C$ (Westerhold et al., 2005) (Fig 5C and 5D). Bottom waters have cooled
by ~2°C and sea surface waters cooled by 6-7°C in the Southern Ocean (Holbourn et al., 2007;
Shevenell et al., 2004), and cooled ~ 2°C of sea surface waters in the Eastern Equatorial Pacific
(Rousselle et al., 2013) (Fig 5F). A 59 ±6 m of sea level fall in northeastern Australia at ~13.9 Ma
occurred due to ice growth on Antarctica (John et al., 2011). Sea level estimates from coastal plain
coreholes in New Jersey and Delaware show an about 40 m fall of sea level between 14 and 11 Ma
(Kominz et al., 2008) (Fig 5E). Increases in opal accumulation from 14 to 13.8 Ma from ODP
U1338 in eastern equatorial Pacific indicated an enhanced siliceous productivity (Holbourn et al.,
2014). During this period, the onset of a perennial sea ice cover in the Arctic Ocean probably
occurred at ~13 Ma (Krylov et al., 2008), and the extinction of tundra in continental Antarctica has
taken place at ~14 Ma (Lewis et al., 2008), and decrease of mass accumulation rates of silicate
sediments occurred at ~15.5 Ma in South China Sea (Wan et al., 2009) (Fig 5G). Some hypotheses
were tried to interpret these paleoclimatic changes, including atmospheric $CO_2$ drawdown
(Holbourn et al., 2005; Shevenell et al., 2008) and orbitally-paced climate changes (Holbourn et
al., 2007). But, the atmospheric $CO_2$ reconstructions still remain unclear (Fig 5B). The eccentricity



may be a pacemaker of middle Miocene climate evolution through the modulation of long-term
carbon budgets (Holbourn et al., 2007) (Fig 5A).

Asian paleoclimate underwent major changes during the middle to late Miocene from

relatively wet interval during ca. 17 to 15 Ma to a more arid one that continued to the present (Hui
et al., 2011; Song et al., 2014). A notable high magnetic susceptibility value interval between 16
and 14 Ma from Zhuanglang site at western Chinese Loess Plateau was interpreted to reflect the
Miocene climatic optimum (Qiang et al., 2011). Sporopollen data from the Tianshui basin at the
NE Tibetan Plateau indicates a dominated temperate, warm-temperate broad-leaved forest
between 17.1 and 14.7 Ma in response to the wet conditions (Hui et al., 2011).

But After ca.15 Ma, dry conditions have prevailed in the inland Asia. Palynological records

from the Tianshui basin (Hui et al., 2011; Liu et al., 2016), Wushan Basin in the Northeastern
Tibetan Plateau (Hui et al., 2017), Guyuan at the Ningxia province (Jiang and Ding, 2008),
western Qaidam basin (Miao et al., 2011), and northern Tian Shan (Tang et al., 2011) show that
the *Artemisia*, *Chenopodiaceae* (Fig 6A), *Ephedra* and *Poaceae* significantly increased and
remained the dominant taxa in the pollen assemblages, indicating a persistent drier condition in
central Asia after the middle Miocene climatic optimum. A rapid decrease of magnetic
susceptibility within the Neogene eolian sequences from the eastern Xorhol basin at the
northeastern Tibetan Plateau indicate that the aridity of Asian interior intensified after 11.5-
10.3 Ma period (Li et al., 2014). Carbonate content from the western Qaidam basin reveal a sharp
decrease since 11 Ma in response to the increase of regional aridity (Song et al., 2014) (Fig 6D).
Isotopic data from pedogenic and lacustrine carbonates in the northeastern Qaidam basin and
Xunhua basin in the northeastern Tibetan Plateau displays a positive shift of ~2.5‰ and ~1.5‰ in
$\delta^{18}O$ values during this period, respectively (Fig 6B), indicating that intensified aridity in central
Asia occurred at~12 Ma (Zhuang et al., 2011; Hough et al., 2014). A similar study from the
southwestern Qaidam basin has shown that a~1.5‰ positive shift in the most negative $\delta^{18}O$ values
of carbonate cements and pedogenic carbonates occurred at 13-12 Ma (Li et al., 2016). Another
similar study from the Qaidam basin show that suddenly decrease of the ostracod species diversity,
abrupt positive shifts of about 3.75‰ in $\delta^{18}O$ values and 5.28‰ in $\delta^{13}C$ values for ostracod
valves, and markedly decrease of the chemical index of weathering (CIW) occurred since 13.3 Ma
ago (Song et al., 2017).  Multiproxies of the Sikouzi section in the Ningxia province in China
changed substantially after 12-11 Ma, with an increase of magnetic susceptibility, lightness and
total inorganic carbon and a decrease of the pollen humidity index, total organic carbon and
redness; these imply that the paleoclimate in central Asia became cooler and drier since 12 Ma
(Jiang et al., 2008). The expansion of the dry areas in western China after ca 15 Ma would supply
a larger amount of the dust to the Lanzhou basin (Zhang et al., 2014) and Chinese Loess Plateau
forming the Red Clay sediments (Xu et al., 2009). The long-term drying of inland Asia after ca 15
Ma led to the disappearance of late Miocene episodic lakes in the Tarim basin and shifted to the
currently prevailing desert environments (Liu et al., 2014). In addition, increased frequencies of



fire in the dry Inner Asia may be related to a continuous aridification in Asia (Miao et al., 2016).
The Asian monsoon apparently changed during 14-8 Ma. Gradually increase percentages of
xerophytic taxa in the Qaidam basin suggest gradual strengthening of East Asian winter monsoon
and weakening of East Asian summer monsoon (Miao et al., 2011). Pollen and grain-size studies
from the Sikouzi area on the east side of the Liupan Mountains also reveal a weak intensity of East
Asian summer monsoon since 12 Ma ago (Jiang and Ding, 2008, 2009). Late Miocene winter
monsoon intensification is evidenced in the decreased magnetic susceptibility variability of
Zhuanglang Red Clay deposits (Qiang et al., 2011) (Fig 6F); which was consistent with the
relatively low calcite/quartz ratios during 9.5-7.5 Ma in response to the strong East Asian winter
monsoon intensity (Sun et al., 2015). Lacustrine micrite and pedogenic carbonate from the
Xunhua basins at the northeastern Tibetan Plateau (Hough et al., 2011, 2014), and from the
northeastern Qaidam basin (Zhuang et al., 2011) show a positive shift of ~1.5‰ and ~2.5‰ in
$\delta^{18}O$ values during this period, respectively, imply an increased regional aridification and related
to enhanced East Asian winter monsoon. Increased mineralogical ratios (chlorite/quartz,
illite/quartz, calcite/quartz and protodolomite/quartz) from the Zhuanglang section in the western
Chinese Loess Plateau indicated weak East Asian summer monsoon intensity during 18.5-9.5 Ma
(Sun et al., 2015). The ratios of (illite+chlorite)/smectite and (quartz+feldspar)% from ODP site
1146 in South China Sea reveal a significant increase at~15 Ma as a result of enhanced winter
monsoon (Wan et al., 2007). The CIA ($100\times Al_2O_3/(Al_2O_3+CaO+Na_2O+K_2O)$) from the same site
1146 show a significant decrease at about 15 Ma related to decreased summer monsoon intensity
(Wan et al., 2009) (Fig 6E). The illite/smectite ratios from IODP U1430 in Japan Sea show a rapid
increase at~11.8 Ma as suggestive of increased eolian input related to enhanced winter monsoon
(Shen et al., 2017) (Fig 6C). A comprehensive review of numerous proxies from the South China
Sea sediments reveals a strong summer monsoon during ~21-18.5 Ma, followed by an extended
period of summer monsoon maximum from 18.5 to 10 Ma, then weakening (Clift et al., 2014).
The South Asian summer monsoon may begin and/or strengthen during this period. The D/H
ratios of pedogenic clay and the $^{18}O/^{16}O$ ratio of carbonate nodules from Siwalik sediments in
India reveal a substantially strengthened Indian monsoon at ~11 Ma (Sanyal et al., 2010). But, the
geophysical and geochemical data from the IODP Expediton 359 in Indian Ocean reveal an abrupt
modern South Asian Monsoon onset at ~12.9 Ma (Betzler et al., 2016), with an apparent decrease
content of Mn/Ca ratios (Fig 6G). This age was also reported by Gupta et al.(2015) based on the
stable isotope analysis of planktonic foraminifera in the Arabian Sea and significant increase of
TOC contents (Fig 6H). Recent research from ODP site 722B and 730A in the western Arabian
Sea revealed a major drop in sea-surface temperature in the period of 11-10 Ma related to the
establishment of monsoonal upwelling (Zhuang et al., 2017).

**2.3.2. Uplifts of the plateau margins linked to this Asian drying**
During this period, the plateau margins underwent rapid uplift again and there was the onset



of S-N rifting in central Tibet (Fig. 7; Locations that mentioned the uplift and deformation at this
part have been shown on the circles and detail information can be seen table 3).

In northeastern Tibet, low-temperature thermochronological and detrital zircon analyses

indicate that the North Qilian Shan (Zheng et al., 2010; Pan et al., 2013; Wang et al., 2016a), Jishi
Shan (Lease et al., 2011), Liupan Shan (Wang et al., 2017), and Haiyuan fault (Duvall et al., 2013)
had undergone accelerated exhumation between 14 and 10 Ma. The rapid deformation and
exhumation of these mountains would lead to hydrologic separation in the adjacent basins, such as
Xunhua and Linxia basins (Hough et al., 2011), and to a high sedimentation rate for foreland
basins and new detrital zircon components (Lease et al., 2012; Liu et al., 2013; Saylor et al., 2017).
A combination of magnetostratigraphy and cosmogenic burial ages from the fluvial deposits in
Gonghe basin, together with lithostratigraphic patterns and paleocurrent records, indicates that the
rise of the Gonghe Nan Shan became significant at ~10 Ma (Craddock et al., 2011). A clockwise
rotation of 25.1±4.6º of the Guide basin took place between 17 and 11 Ma (Yan et al., 2006). A
magnetostratigraphic study of the Dahonggou section in the northern Qaidam basin coupled with
the variations in lithofacies, sedimentation rate and magnetic susceptibility reveal that the Qilian
Shan and the Altyn Tagh fault were synchronously tectonically active at ~12 Ma (Lu and Xiong,
2009). This time was consistent with the onset of molasse deposits along the Altyn Tagh fault at
about 13 Ma (Sun et al., 2005). In the northwestern Tibet, apatite fission track results reveal that
the West Kunlun range experienced rapid cooling and exhumation during 12-8 Ma, which was
consistent with sharply increased sedimentation rates at the southern margin of the Tarim basin
(Wang et al., 2003). The uplift and erosion of the Tian Shan accelerated at ~11 Ma, as constrained
by a two-fold increase in sedimentation rate as well as marked changes in rock magnetic
characteristics at this time in the Yaha section on the southern flank of the central Tian Shan
(Charreau et al., 2006).

In Himalayas, the extrusion rate of the Higher Himalayan Crystalline thrust sheet onto the

Lesser Himalaya sequence slowed in the middle Miocene and ceased by ca. 12 Ma (Godin et al.,
2006). The activity of the Main Central thrust and the South Tibetan Detachment System had
ceased by 13-12 Ma based on U-Pb ages of deformed pegmatites, $^{40}$Ar/$^{39}$Ar hornblende ages and
Rb-Sr cooling ages of muscovite and biotite (Catlos et al., 2002; Daniel et al., 2003; Tobgay et al.,
2012). The Main Boundary thrust began active during 12-9.5 Ma inferred from the regional
increasing erosion in the Lesser Himalaya and rates of the foreland-basin fill (Huyghe et al., 2001;
Meigs et al., 1995). Thiede et al. (2009) integrated 255 apatite and zircon fission track and white
mica $^{40}$Ar/$^{39}$Ar ages from the northwest Himalaya, and suggested that a high exhumation rate of 1-
2 mm/a existed since 11 Ma along the southern High Himalayan slopes. In the Tethyan Himalaya,
the rapid exhumation range was from 17 to 5.7 Ma in the central Himalaya and from 15 to 3 Ma in
the southwestern Himalaya (Liu et al., 2005; Thiede et al., 2005). A series of N-S striking rifts and
high-angle normal faults were documented in the Himalaya, such as the Kung Co, Thakkola,
Yadong-Gulu. Based on magnetostratigraphy of the Tetang Formation, the initiation of Thakkola



rift extension was constrained between 11 and 10 Ma (Garzione et al., 2000, 2003). The zircon and
apatite (U-Th)/He ages from the footwall of the early Miocene Kung Co granite in southern Tibet
suggest that initiation of normal fault slip was at ~13-12 Ma and that rapid exhumation of the
footwall was between ~13 Ma and 10 Ma (Lee et al., 2011). In eastern Tibet, low-temperature
thermochronological results show that southwestern Longmen Shan experienced rapid cooling at
15 Ma (Cook et al., 2013), the central Longmen Shan was initially active at ~11 Ma (Kirby et al.,
2002), and the northeastern part of Min Shan was at 7-4 Ma (Kirby et al., 2002). Moreover, the
thermochronlogic analyses from the central and southern Longmen Shan Thrust-Nappe belt reveal
differential cooling across the Erwangmiao and Yingxiu-Beichuan faults during Miocene (Arne et
al., 1997).

We cannot rule out the effects of global cooling during this period, which would reduce the

amount of water vapor held in the atmosphere and thereby can cause terrestrial drying. But, the
further outward-growth of the plateau margins played an important role for Asian drying. First,
Miao et al.(2012) examined the evolution of Miocene climate for five separate regions in Eurasia,
including Europe, High-latitude Asia, the East Asian Monsoon region, the South Asian Monsoon
region, and Central Asia. The results show that the moisture evolution in Central Asia shows less
similarity with other four regions, and thereby the uplift of the plateau margins could provide a
possible explanation for these differences. Second, climatic proxies from the Central Asia, Japan
Sea and South China Sea (Fig.6) do not show synchronously changes in response to global cooling.
If we do not consider the age reliable, this may imply that regional factors, especially differential
uplift of the marginal mountains on the edge of the Tibetan Plateau, played an important role for
proxy changes in the context of Middle-Late Miocene global cooling.

## 3. Discussion

At least four hypotheses are proposed to interpret the Asian aridification changes: (1) the

uplift of the Tibetan Plateau (e.g., Miao et al., 2012; Zheng et al., 2015); (2) the retreat of the
Tethys Sea in Asia (e.g., Bosboom et al., 2014a; Ramstein et al., 1997); (3) the global cooling
during the Cenozoic (e.g., Dupont-Nivet et al., 2007; Lu and Guo, 2013); and (4) the decreasing
concentration of atmospheric $CO_2$ (e.g., Lu and Guo, 2013). Previous studies show that the retreat
of the Tethys Sea occurred around 47-40 Ma. This regression was coeval with the initial
aridification of the central Asia, the regional disappearance of a relatively wet perennial saline
lake system, and a prominent shift to relatively more arid flora around ~41 Ma recorded in the
Xining basin (Bosboom et al., 2014a; Sun et al., 2016). Therefore, some scholars suggested that
the sea retreat in central Asia played an important role in the deterioration of the Asian
paleoenvironment (Bosboom et al., 2014b; Ramstein et al., 1997). The global cooling is another
factor for Asian desertification. The cooling would not only cause ice-sheet expansion and an
increase in meridional temperature gradients leading to the southward retreat of summer monsoon,



but also would reduce the amount of water vapor held in the atmosphere leading to both additional
cooling and further weakening of the East-Asian summer monsoon (e.g., Dupont-Nivet et al.,
2007; Jiang and Ding, 2008; Lu and Guo, 2013). The decrease in average atmospheric $CO_2$
concentration would not only cause global cooling but also would shift the inter-tropical
convergence zone southward, thereby reducing the monsoon precipitation accompanied by the
intensification of Asian desertification (e.g., Anagnostou et al., 2016; Lu and Guo, 2013).

Although numerous elements influence evolution of East Asian climate, we consider that the

three main phases of uplift of the Tibetan Plateau region played an important role in drying in Asia.

During the first pulse, the central Tibet reached the near-modern elevation and probably the

Himalayas had already obtained the present-day elevation by at least 35 Ma. The high elevation in
central Tibet would increase silicate weathering and erosion contributing to lowering of
atmospheric $CO_2$, which was a major cause of global cooling (e.g., Dupont-Nivet et al., 2008a;
Garzione, 2008). Global deep-sea oxygen records show a significantly positive shift in response to
rapid global cooling during 50-35 Ma (Fig. 8). Reconstructions of atmospheric $CO_2$ concentrations
based on the boron isotope composition of well preserved planktonic foraminifera show a relative
decline in $CO_2$ concentrations through the Eocene of about 50 ppm that would be sufficient to
drive the high-and low-latitude cooling during late Eocene (e.g., Anagnostou et al., 2016). The
continuing uplift of the plateau, combined with a decrease of seafloor spreading rates, would result
in declining atmospheric $CO_2$ concentrations below ~760 ppm allowed for a critical expansion of
ice sheets on Antarctica (Dupont-Nivet et al., 2008a; Pearson et al., 2009). In addition, the
continuing northward injection of the Pamir related to the Tibet uplift forced the long-term
westward sea retreat from the Tarim basin (Carrapa et al., 2015; Sun et al., 2016). This resulted in
the regional initiation of the Asian aridification induced by the decrease of moisture supplied from
the Paratethys Sea (Bosboom et al., 2014b). More notably, the high Himalayas and south Tibet
would lead to the formation of the south Asian monsoon by orographic insulation (Boos and
Kuang, 2010) or thermal forcing (e.g., Wu et al., 2012). However, the warm and moist air from the
Indian Ocean could not easily flow toward the central and northern Tibet due to the topographic
barrier of the high Himalayas. Additionally, the significant uplift of the northern Tibet during this
interval probably caused a relatively weak monsoon-like climate during the Eocene time, which
was consistent with recent climate model simulations that the uplift of northern Tibet was critical
for intensification of East Asian monsoon (Liu and Dong, 2013; Liu et al., 2015a; Tang et al.,

2013).

The second pulse between 30 and 20 Ma is characterized by a further uplift of the plateau

margins. However, the intense uplift of the plateau margins during this period cannot interpret the
rapid warming of global climate during the Late Oligocene, which suggests that the process of
silicate weathering of these elevated mountain belts and the subsequent sequestration of carbon
was not sufficient in itself to counter the recorded relative rise in atmospheric $CO_2$ concentration
(Fig. 8). Instead, this late-Oligocene climatic warming may have been partly a side-effect of a



decrease of organic carbon burial and a net addition of $CO_2$ to the atmosphere (e.g., Raymo and
Ruddiman, 1992). Nevertheless, the uplift of the plateau margins during this interval had major
regional impacts on the climate of central Asia. Climatic simulations reveal that the uplift of the
northern Tibet would cause an initial formation of the East Asian monsoon as well as the
desertification in central Asia (e.g., Liu and Dong, 2013; Liu et al., 2015a, 2017; Zhang et al.,
2012, 2017). Moreover, the intense uplift of the northern margins would strongly strengthen the
land-sea thermal contrast, thereby leading to intensification of the East Asian winter monsoon and
reducing precipitation in inland Asia (Wu et al., 2012). The synchronous occurrence of the plateau
uplift and intensification of the East Asian monsoon suggests that the uplift of the plateau margins
was the primary mechanism for the climatic variations in central Asia during this period.

The third uplift of the plateau from 15 to 8 Ma was dominated by the uplift of the plateau

margins. Although Willenbring and von Blanckenburg (2010) suggested that pulses in mountain
uplift over the past 10 Ma might have been neither a direct cause nor an inevitable consequence of
climate change, we consider that the Asian drying changes during this interval are primarily
attributed to the rapid uplift of the Tibetan Plateau coupled with the global cooling (Fig. 8).
Temperature and moisture proxy data from the five regions (Europe, high-latitude Asia, East Asian
monsoon region, South Asian monsoon region, and Central Asia) suggests that the moisture
evolution of central Asia was largely decoupled from adjacent regional trends during the mid-late
Miocene, implying that the uplift of the Tibetan Plateau played an important role in the
strengthening of aridification in central Asia (Miao et al., 2012). Climatic simulations show that
the uplift of the northern Tibet would enhance the desertification of inland Asia and
simultaneously strengthen the East Asian winter monsoon (Liu et al., 2015a; Tang et al., 2013).
There is some evidence of a significant weaken of the East Asian summer monsoon from 14 to 11
Ma. But Chemical weathering data from ODP site 1146 and 1148 in South China Sea suggests that
the summer monsoon was relatively constant and wet during 14-10 Ma (Clift et al., 2008, 2014).
After 11 Ma, the further strengthening of East Asian winter monsoon was attributed to the
Himalaya-Tibetan Plateau uplift and global cooling (e.g., An et al., 2001).

Although we try to establish the linkages between the uplift of the Tibetan Plateau and Asian

climatic evolution, the effects between global cooling and the Tibetan Plateau uplift can still not
be differentiated. Climate models did not take into account the detailed topography and other
boundary conditions at each stage of the uplift (Tada et al., 2016). Additionally, there are still
widely debates on paleoaltimetry of the Tibetan Plateau (Deng and Ding, 2015). Thus, more
accurate evolution of the Tibetan Plateau uplift and the paleoclimatic variations in Asia should be
reestablished in future.

## 4. Conclusion


The growth stages of the Tibetan Plateau and its margins during the Cenozoic had a series of

potential effects on Asian climate. During the first stage (~55-35 Ma; Eocene), the central Tibet



has obtained near-modern elevation accompanied by the significant uplift of the northern margins.
The high elevation of south Tibet would increase rates of silicate weathering, thereby leading to
the drawdown of atmospheric $CO_2$ and contributing to global cooling. Meanwhile, the progressive
northward trend in uplift of the plateau probably forced the long-term westward withdrawal of the
Paratethys Sea, which contributed to the onset of regional Asian desertification by decreasing
moisture supply. The global cooling and sea retreat, coupled with the topographic barrier effect of
the Tibetan Plateau, were major factors in the initial aridification of central Asia.
The second uplift stage during late Oligocene and early Miocene is characterized by
relatively little tectonic activity in central Tibet, but by a further uplift of the plateau margins. The
uplift of northern margin of Tibet during this interval led to the onset of East Asian winter
monsoon as well as the intensive desertification of inland Asia. During the third stage, from 15 to
8 Ma, the plateau margins again underwent major uplift, thereby further strengthening the Asian
winter monsoon and the desertification of the inland Asia.

*Acknowledgments: we are grateful to Jim Ogg for language editing that notably improved the*
*manuscript. This work was supported by Natural Science Foundation for Distinguished Young*
*Scholars of Hubei Province of China (2016CFA051) and the National Natural Science Foundation*
*of China (No. 41322013).*

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



Fig. 7. Topographic map of the Tibetan Plateau showing evidence of rejuvenation or initiation of tectonic activities
during 15-8 Ma. The black circles represent some geographic locations mentioned in the article.The detailed
information of tectonic activities during 15-8 Ma is shown in table.3.

Fig 8. Evolution of Asian climate and the Tibetan Plateau, and their relation with global changes during the
Cenozoic. The data of benthic foraminiferal $\delta^{18}O$ and atmospheric $CO_2$ content is modified from Zachos et al.
(2001) and Zachos et al. (2008), respectively.


Figure 1

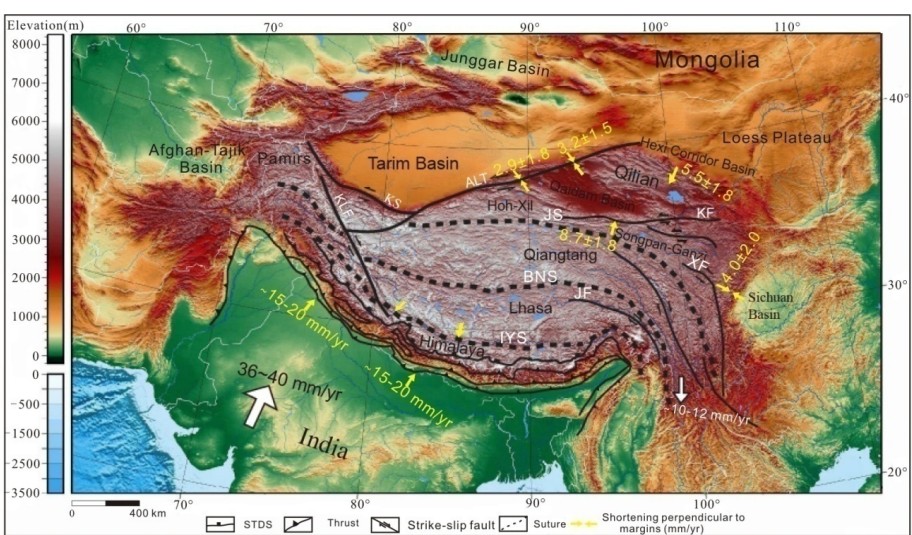




















Figure 2

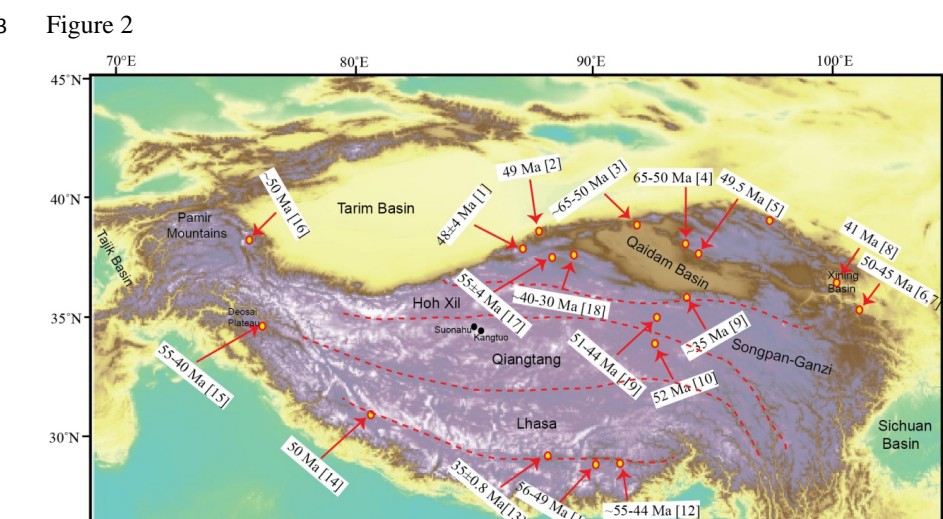




























Figure 3

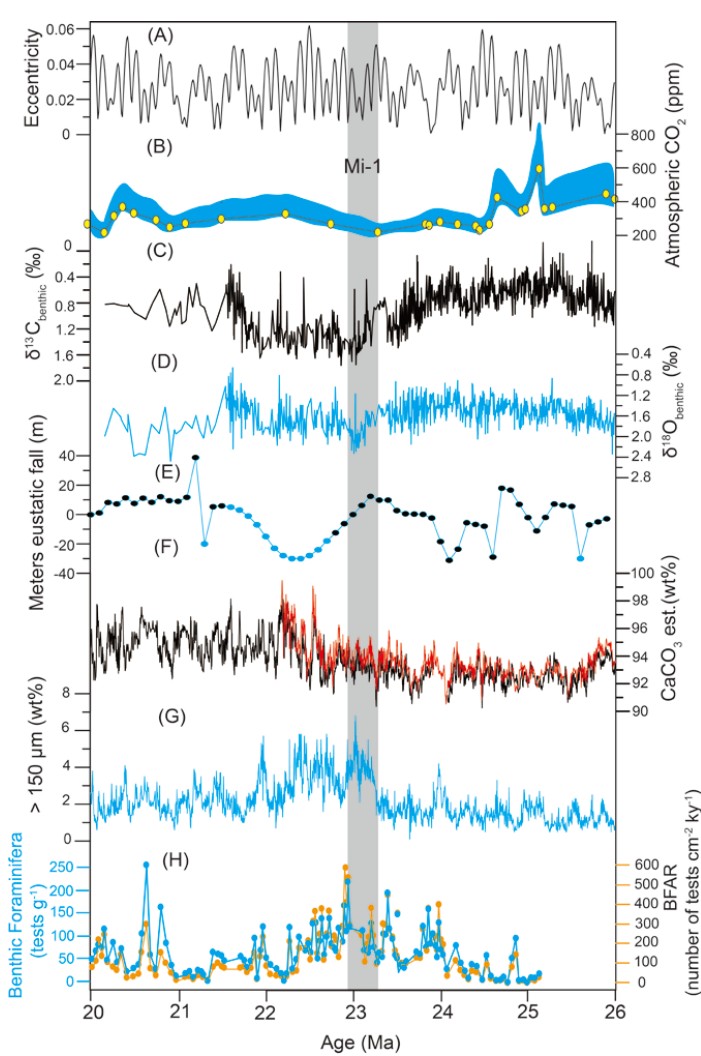















Figure 4

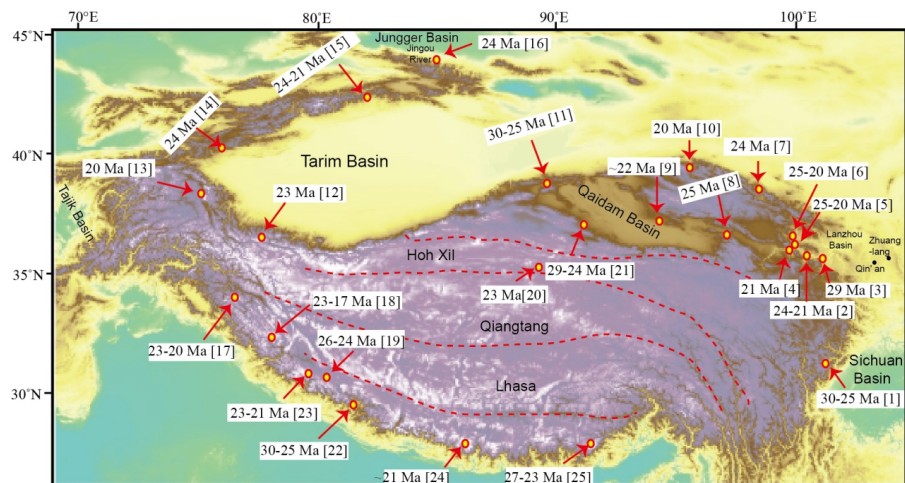




























Figure 5

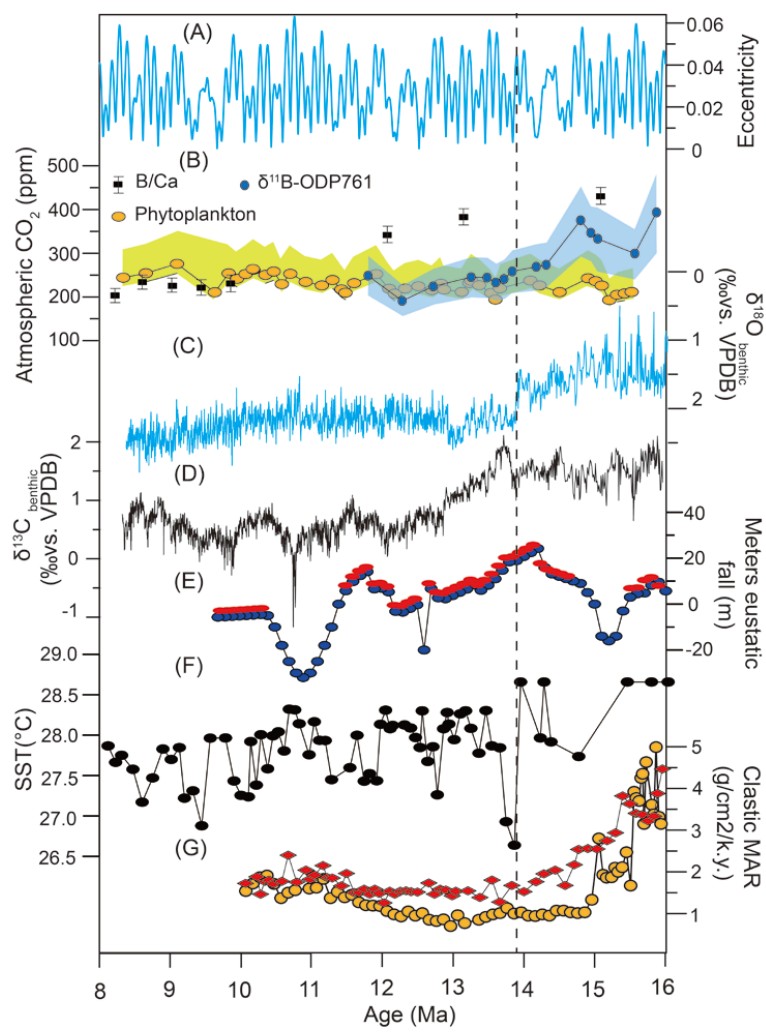
















Figure 6

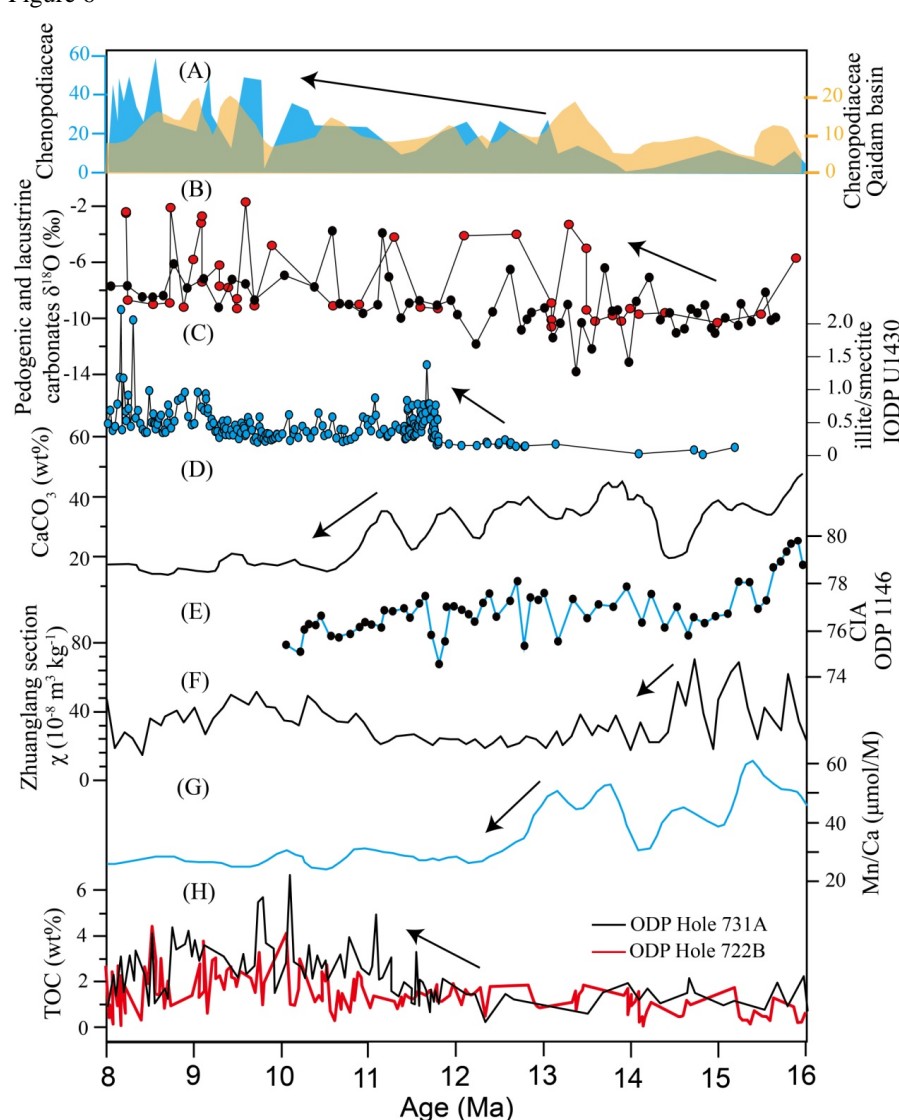














Figure 7

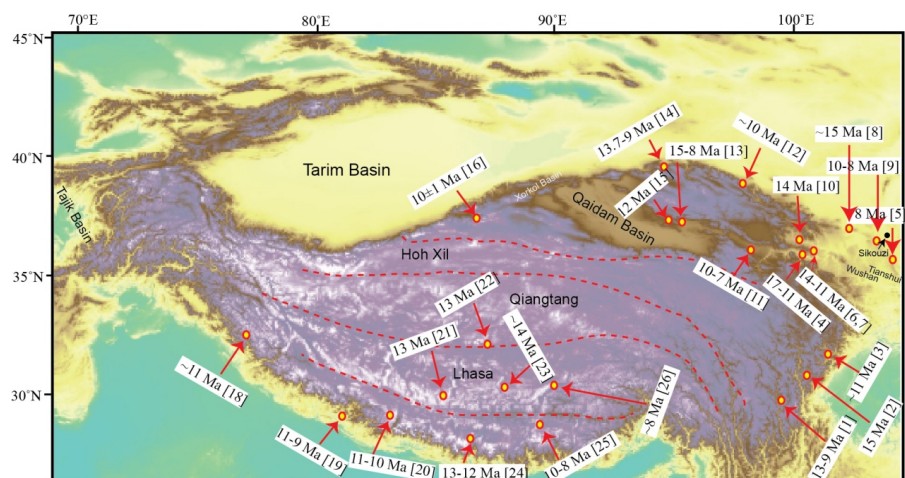




Figure 8

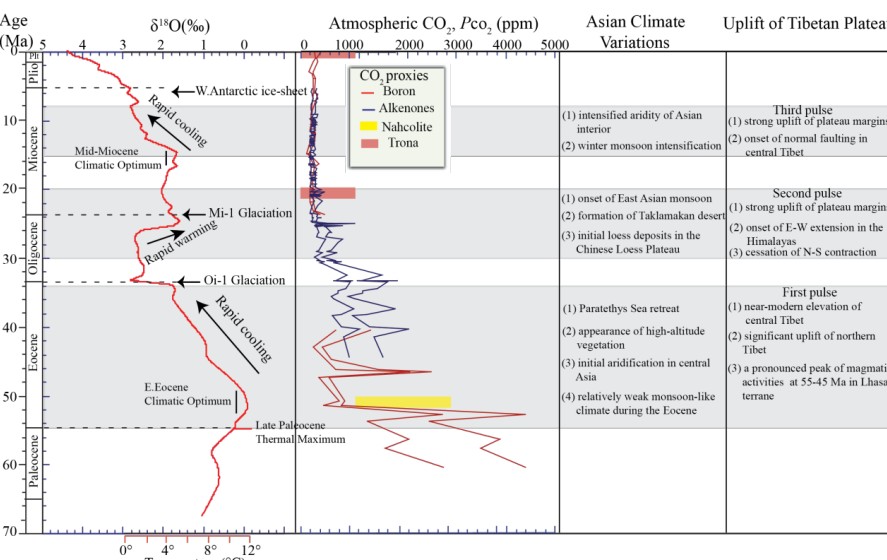




1367                                    **Table captions**

Table 1. Detailed information of rejuvenation or initiation of tectonic activities at ~65-
35 Ma in the Tibetan Plateau.




Table 2. Detailed information of rejuvenation or initiation of tectonic activities at 30-
20 Ma in the Tibetan Plateau.

Table 3. Detailed information of rejuvenation or initiation of tectonic activities at 15-8
Ma in the Tibetan Plateau.

Table 1

| Region/thrusts | Events and Ages | Methods | References | Marks |
|---|---|---|---|---|
| Altyn Tagh fault | Initial active (~49 Ma) | Apatite fission track (AFT) | Jolivet et al. (2001),Yin et al. (2002) | [1,2] |
| Altyn Tagh | Rapid uplift (65-50 Ma) | Zircon fission track and sedimentary | Wang et al. (2015) | [3] |
| Northern Qaidam basin | Initial contraction (65-50 Ma) | Seismic reflection and balanced cross section | Yin et al. (2008a) | [4] |
| Northern Qaidam basin | High accumulation rate (~49.5 Ma) | Sediments | Wei et al. (2013) | [5] |
| West Qinling thrust | Rapid cooling (50-45 Ma) | Apatite (U-Th)/He | Clark et al. (2010) | [6] |
| West Qinling thrust | Initial active (~50 Ma) | $^{40}Ar$-$^{39}Ar$ | Duvall et al. (2011) | [7] |
| Xining basin | 25°of clockwise rotation (41 Ma) | Paleomagnetic data | Dupont-Nivet et al. (2008b) | [8] |
| Eastern Kunlun | Rapid cooling (~35 Ma) | Apatite (U-Th)/He | Clark et al. (2010) | [9] |
| Tanggula thrust | Initial active (~52 Ma) | Sediments and angular unconformities | Li et al. (2012) | [10] |
| Tethyan Himalaya thrust | Initial active (56-49 Ma) | $^{40}Ar$-$^{39}Ar$ | Wiesmayr and Grasemann (2002) | [11] |
| Tethyan Himalayan | Significant crustal thickening(~55-44 Ma) | U-Pb, K-Ar and $^{40}Ar$-$^{39}Ar$ | Aikman et al. (2008) | [12] |
| Mabja Dome | Onset of mid-crustal extension (35±0.8 Ma) | U-Pb | Lee and Whitehouse (2007) | [13] |
| Northwestern Himalaya | Large-scale granite intrusion (~50 Ma) | U-Pb | Wang et al. (2012a) | [14] |
| Deosai plateau | Rapid cooling (55-40 Ma) | AFT, apatite and zircon (U-Th)/He | van der Beek et al. (2009) | [15] |
| Kashgar-Yecheng thrust | Initial motion (~50 Ma) | zircon and apatite fission track | Cao et al. (2013) | [16] |
| Kunlun fault | Rapid cooling (55±4 Ma) | AFT | Jolivet et al. (2001) | [17] |
| Qimen Tagh mountains | Initial uplift (~40-30 Ma) | AFT | Liu et al. (2015b) | [18] |
| Fenghuoshan fold-thrust belt | Initial deformation (51-44 Ma) | AFT, apatite (U-Th)/He, $^{40}Ar$-$^{39}Ar$ | Staisch et al. (2016) | [19] |






Table 2

| Region/thrust | Events and Ages | Method | Reference | Marks |
|---|---|---|---|---|
| Longmen Shan | Initial uplift (30-25 Ma) | AFT, apatite and zircon (U-Th)/He | Wang et al. (2012b) | [1] |
| Xunhua basin | High accumulation rates (24-21 Ma) | Magnetostratigraphy | Lease et al. (2012) | [2] |
| Linxia Basin | Rapid subsidence (29 Ma) | Magnetostratigraphy | Fang et al. (2003) | [3] |
| Guide Basin | Sedimentary discontinuity (21 Ma) | Structural geology and sedimentary | Liu et al. (2013) | [4] |
| Laji Shan | Rapid uplift (25-20 Ma) | Magnetostratigraphy and U-Pb | Lease et al. (2012) | [5] |
| Xining basin | Unstable accumulations (25-20 Ma) | Magnetostratigraphy | Xiao et al. (2012) | [6] |
| Northeast Qilian | Rapid cooling (24 Ma) | AFT | Pan et al. (2013) | [7] |
| Elashan | Rapid uplift (25 Ma) | Magnetostratigraphy and AFT | Lu et al. (2012) | [8] |
| North Qaidam basin | Fault reactivation(~22 Ma) | Constrained by the sedimentary | Lu and Xiong (2009) | [9] |
| North Qilian | Rapid exhumation (20 Ma) | AFT, vitrinite-reflectance analysis | George et al. (2001) | [10] |
| Altyn Tagh fault | Rapid exhumation (30-25 Ma) | AFT and sediments | Jolivet et al. (2001) | [11] |
| West Kunlun Shan | Rapid uplift (23 Ma) | Seismic reflection and drill-well data | Jiang et al. (2013) | [12] |
| Main Pamir thrust | Rapid cooling (20 Ma) | AFT | Sobel and Dumitru (1997) | [13] |
| Southwest Tian Shan | Rapid exhumation (24 Ma) | AFT | Sobel et al. (2006) | [14] |
| Southern Tian Shan | Initial uplift (24-21 Ma) | Sedimentary record and Magnetostratigraphy | Yin et al. (1998) | [15] |
| Northern Tian Shan | Initial unroofing (~24 Ma) | AFT | Hendrix et al. (1994) | [16] |
| Zanskar Shear Zone | Cooling ages of muscovites (23-20 Ma) | $^{40}$Ar-$^{39}$Ar | Walker et al. (1999) | [17] |
| Sutlej Rift | Exhumation of deep crustal rocks (23-17 Ma) | $^{40}$Ar-$^{39}$Ar | Vannay et al. (2004) | [18] |
| Kailas basin | Initial deposition (26-24 Ma) | Igneous zircon U-Pb age | DeCelles et al. (2011) | [19] |
| Hoh Xil basin | Sedimentary discontinuity (23 Ma) | Constrained by sedimentation | Wang et al. (2002) | [20] |
| Eastern Kunlun | Initial uplift (29-24 Ma) | Constrained by sedimentary of foreland basin | Yin et al. (2008b) | [21] |
| Ama Drime range | Partial melting (30-25 Ma) | (U-Th)/He, $^{40}$Ar-$^{39}$Ar and U-Th/Pb | Kali et al. (2010) | [22] |
| Silving leucogranite | Rapid cooling (23-21 Ma) | U-Pb, AFT and $^{40}$Ar-$^{39}$Ar | Searle et al. (1999) | [23] |
| Everest | Initial movement of MCT (~21 Ma) | U-Pb and $^{40}$Ar-$^{39}$Ar | Viskupic et al. (2005) | [24] |
| Gangdese thrust | Initial motion (27-23 Ma) | $^{40}$Ar-$^{39}$Ar | Yin et al. (1994) | [25] |










Table 3

| Region/thrust | Events and Ages | Method | Method | Marks |
|---|---|---|---|---|
| Southeastern Tibet | Rapid cooling (13-9 Ma) | AFT and apatite (U-Th)/He | Clark et al. (2005b) | [1] |
| Southwestern Longmen | Rapid exhumation (15 Ma) | AFT, apatite and zircon (U-Th)/He | Cook et al. (2013) | [2] |
| Central Longmen Shan | Rapid exhumation ( ~11 Ma) | $^{40}$Ar-$^{39}$Ar, apatite and zircon (U-Th)/He | Kirby et al. (2002) | [3] |
| Guide basin | Clockwise rotation (17-11 Ma) | Paleomagnetic data | Yan et al. (2006) | [4] |
| Liupan Shan | Rapid exhumation (~8 Ma) | AFT | Zheng et al. (2006) | [5] |
| Jishi Shan | Rapid uplift (14-11 Ma) | AFT, apatite (U-Th)/He and U-Pb | Lease et al. (2011,2012) | [6,7] |
| Central Haiyuan fault | Initial motion (~15 Ma) | AFT, apatite and zircon (U-Th)/He | Duvall et al. (2013) | [8] |
| Eastern Haiyuan fault | Initial motion (10-8 Ma) | AFT, apatite and zircon (U-Th)/He | Duvall et al. (2013) | [9] |
| Xining basin | Significant uplift (14 Ma) | Paleomagnetic age of river terraces | Lu et al. (2004) | [10] |
| Gonghe Nan Shan | Initial active (10-7 Ma) | Constrained by sediments of foreland basin | Craddock et al. (2011) | [11] |
| North Qilian Shan | Rapid cooling (~10 Ma) | Apatite (U-Th)/He | Zheng et al. (2010) | [12] |
| Eastern Qaidam basin | High accumulation rates (15-8 Ma) | Inferred from magnetostratigaphy | Fang et al. (2007) | [13] |
| Altyn Tagh | Rapid uplift (13.7-9 Ma) | Paleomagnetic age of molasse deposits | Sun et al. (2005) | [14] |
| Southern Qilian Shan | Rapid uplift (12 Ma) | Magnetostratigraphy | Lu and Xiong (2009) | [15] |
| Altyn Tagh fault | Rapid cooling (10±1 Ma) | AFT | Jolivet et al. (2001) | [16] |
| West Kunlun | Rapid uplift (12-8 Ma) | AFT | Wang et al. (2003) | [17] |
| Sutlej Valley | Peak metamorphism (~11 Ma) | U-Pb ages | Caddick et al. (2007) | [18] |
| Main Boundary thrust | Initial motion (11-9 Ma) | Inferred from sediments in Siwalik Group | Meigs et al. (1995) | [19] |
| Thakkola rift | Initial extension (11-10 Ma) | Magnetostratigraphy | Garzione et al. (2000) | [20] |
| Tangra Yumco rift | Initial extension (13-12 Ma) | Zircon and apatite (U-Th)/He | Dewane et al. (2006) | [21] |
| Shuanghu rift | Initial extension (13 Ma) | $^{40}$Ar-$^{39}$Ar | Blisniuk et al. (2001) | [22] |
| Xainza rift | Initial extension (~14 Ma) | U-Pb and apatite (U-Th)/He | Hager et al. (2009) | [23] |
| Kung Co rift | Initial extension (13-12 Ma) | Zircon and apatite (U-Th)/He | Lee et al. (2011) | [24] |
| Yadong-Gulu rift | Initial extension (10-8 Ma) | Constrained by monazite Th-Pb ages | Edwards and Harrison (1997) | [25] |
| Nyainqentanglha rift | Initial extension (~8 Ma) | $^{40}$Ar-$^{39}$Ar | Harrison et al. (1995) | [26] |
