# Peer review of "Three main stages in the uplift of the Tibetan Plateau during the"

_Climate of the Past, 2018_

## Referee Comment (RC1) · PhD Ingalls (Referee) · 26 Jul 2018

Review of "Three main stages in the uplift of the Tibetan Plateau during the Cenozoic period and its possible effects on Asian aridification: A review" by Wang et al.

Overview Based on the scores I have given to the "principal criteria" of Climate of the Past (below), I do not present a comprehensive review of this work. I present rationale for the scores I have given, major comments and suggestions for improvement, and address the review questions provided by CP, as well as a few additional comments.

Journal-specific criteria: Scientific significance: 3 Overall, while this manuscript con-

solidates a great deal of results and findings from the past decade or so on the topic of Tibetan uplift as related to global and local climate change, I do not think a considerable amount of new information or thought is presented in this work. Scientific quality: 3 There was little critical analysis of the data from other studies, so I do not have much to say about scientific quality. Presentation quality: 2 The acknowledgments suggest this manuscript has been edited for English language, but it needs more work before it is ready for publication. The overall structure of the manuscript is fine.

Major comments Insufficient background on the analytical techniques used in the cited works: Line 428 is an example of a common theme in this work in which the authors relay data from a previous study and state what the data imply, but lack additional background necessary for the reader to understand. In this specific example, the authors cite a "positive shift" in oxygen isotopic values and say that these shifts "imply an increased regional aridification and related to enhanced East Asian winter monsoon." However, no where in the work do they explain how oxygen isotopes are related to aridity or how they can be used to make inferences about atmospheric circulation and weather patterns.

Incorrect use of jargon with respect to stable isotopes: I cannot speak to the discussion of paleomagnetism and radiogenic isotope techniques in this work, but I would caution some of the language used with respect to stable isotopes. The authors say "more positive/negative" or "positive/negative shifts" multiple times. A value is positive or negative and cannot be more or less positive or negative. A molecule can have a lighter/heavier isotopic composition w.r.t. a specific isotope/element, or have a lower or higher value. This may seem like a small thing, but will unecessarily irk some readers.

Synthesis: The discussion section mostly summarized everything outlined in Section 2 without much additional discussion of the data or contradicting studies. I think for this work to be useful for the community, it should include a more substantive addition to the discourse rather than just a fairly comprehensive laundry list of recent results. Further, the last paragraph of the Discussion calls into question what has come in light

of the authors' study of the recent literature. Lines 594-596 state that the authors could not draw linkages between the uplift of the TP and evolution of Asia's climate, which seems to be the motivation of the entire study. If this is true, what has been learned? In the same paragraph, the authors say that climate models do not take into account "detailed topography", but in addition to other such climate modeling work, the authors cite multiple studies that use topographic boundary conditions to constrain the effect of TP uplift on global and regional climate (as recent as the previous paragraph even). I think the Discussion section would be much improved if this paragraph was removed and replaced with a synthesis of the use of different types of proxies in each of the three tectonic intervals: which proxies seem to agree between the intervals? Which work best and which have greater uncertainty? If the authors believe more studies are needed on topographic boundary conditions, during which intervals and in which sedimentary basins and/or orogenies? These types of questions and answers can help guide the community, which is the ultimate goal of a review paper.

Figures Figures 1, 2, and 4: It is useful to see geographically and from what tectonic domains the data used in your interpretations of "rejuvenation or initiation of tectonic activities" comes from, but because different proxies were used in each of the studies marked on the maps, and each proxy records a different thermal regime/extension/rotation/magnetism/etc., it's unclear to me how the different points on the map can be related by the viewer. This ties back into my overall comment that the reader needs more background on the commonly used techniques in many of your cited studies to assess what each proxy actually records under the umbrella of "rejuvenation of tectonic activities".

Figure 8: It would be interesting to see some of the additional proxies plotted in Figures 3, 5, and 6 throughout the span of the Cenozoic along with the benthic foraminifera oxygen isotopic composition. The oxygen curve in Figure 8 has been replicated and discussed in numerous studies since its original publication by Zachos, so it would be more intriguing to see how the other proxies change or do not change during the three

pulses you attribute climate change and aridification to. Do pedogenic carbonate d18O and wt. % CaCO3 also follow the benthic foram ïA̧d'18O curve?

CP Reviewer Questions 1. Does the paper address relevant scientific questions within the scope of CP? Yes. 2. Does the paper present novel concepts, ideas, tools, or data? No. 3. Are substantial conclusions reached? No. 4. Are the scientific methods and assumptions valid and clearly outlined? The hypothesis is that aridity is caused by three tectonic "pulses" of uplift and activation on the TP. The authors use data and data interpretation from previous studies to support this hypothesis. However, I do not think enough background on the reviewed works and analytical tools therein is presented for a reader to critically review the summation of data and interpretation the authors of this work come to. 5. Are the results sufficient to support the interpretations and conclusions? See previous comment. 6. Is the description of experiments and calculations sufficiently complete and precise to allow their reproduction by fellow scientists (traceability of results)? N/A 7. Do the authors give proper credit to related work and clearly indicate their own new/original contribution? The authors provide the appropriate citations for the data and data interpretations they summarize. The authors' scientific contribution is the synthesis of this 8. Does the title clearly reflect the contents of the paper? Yes. 9. Does the abstract provide a concise and complete summary? Yes. 10. Is the overall presentation well structured and clear? Yes. 11. Is the language fluent and precise? This paper needs further editing for English usage. 12. Are mathematical formulae, symbols, abbreviations, and units correctly defined and used? N/A 13. Should any parts of the paper (text, formulae, figures, tables) be clarified, reduced, combined, or eliminated? This is not a critical comment, but I think the map figures could be made into one figure with the three age ranges denoted by colored labels. 14. Are the number and quality of references appropriate? There are several instances where the authors choose to cite one paper when there are numerous papers that have contributed to the finding that they are referencing. In some cases it may be that the author is unaware of current and recent work, as it is hard to keep up with everything going on in the TP community, but other instances seem to be
preferential choosing to make a point but may be unfounded. For example, in line 598, the authors cite Deng and Ding (2015) to say that the past elevations of the TP are still debated, which is true, but this paper is not the most recent work to be done and many advances have been made in the past 3 years. It comes off as either an arbitrary or an overly selective choice of citation. 15. Is the amount and quality of supplementary material appropriate? There is no supplement as far as I am aware.

––––––––––––––––––––––––––––

---

## Referee Comment (RC2) · Anonymous Referee #2 · 18 Aug 2018

Scientific significance: 4 The authors do not provide new data, analysis, or concepts. The links between silicate weathering and CO2 drawdown have been with us for multiple decades. The discussion of links between tectonic events on the Tibetan Plateau and climate events cherry-picks events and muddles their timing.

Scientific quality: 3 See comments above.

Presentation quality: 3 Figures are generally acceptable. The English needs editing. I have highlighted some, but not all, of the problematic areas in the detailed comments below.

Major comments The authors need to clearly delineate time periods of interest. For

example, currently all middle–late Miocene climate changes are lumped together, even though the original authors discriminate different mechanisms for climate changes within this period. Similarly, the Cretaceous paleoelevation history of the plateau is largely ignored. See comment on line 400 for an example. Documented uplift events and paleoelevation are not clearly correlative to climate shifts. For example see comment on lines 537–542. Data and conclusions are apparently reported largely without context or comment. This may be fine for an annotated bibliography, but for a review paper, some context and analysis of claims is needed. Otherwise, readers might as well go read all of the citations for themselves. The lack of critical analysis of the data obviates the need for this review. See comments on lines 68 or 397 for an example. Alternative explanations for the observed climate change are dismissed out of hand, without presentation of counter-evidence.

Detailed comments Line 15: Delete "the" before "Tibet" Line 16, 28, 517: Delete "the" before "uplift" Line 19, 26, 77, 277, 343: Replace "during" with "from" Line 20: Delete "n" in "Himalayan" Line 23: Delete "the" before "Eocene" Line 23–24: Delete "the" before "northern" Line 25: Delete "the" before "central" Line 39: The Cenozoic is an era. Line 57: Why "was interpreted"? Reconsider verb tense. Line 66: Add "the" before "Lhasa" Line 68: What about the detrital zircon geochronology suggests that Indo-Asian collision occurred at that time? The authors need to provide sufficient detail for readers to evaluate the claims without having to read the cited literature in its entirety. Line 69–72: This sentence does not make sense. Rewrite. Line 80: Replace "activities" with "activity" Line 82: Awkward. Rewrite. Line 83: No caps on "Paleomagnetic" Line 86: What paleomagnetic results? See comment on line 68. Line 94: Begin this sentence with, "The second is…." Line 111: What "simulated" data? Line 115: Why "continued"? Is the plateau still being uplifted? Line 121–124: This is not accurate. In the models of Boos and Kuang, the Himalaya do not act as a heat pump, but rather act as a barrier which prevents cold and dry Asian air masses from penetrating southward into the Indian subcontinent. Line 139: This whole line of argumentation ignores recent evidence for high elevations in southern Tibet by the Late Cretaceous (e.g., Ding et al.,

2014). Line 143: Replace "by providing" with "due to" Line 152: References? Lines 157–158: Basins and the Monsoon cannot have fossil leaf trait spectra. Rewrite. Line 180: Delete "second" Line 189: DeCelles et al. (2002) is not an appropriate reference here. Line 195: van Hinsbergen et al. (2011) do not present any balanced cross-sections. Line 230: Change "offer a large amount for" to "would provide abundant". Line 236: Delete "the" before "thermal" Line 237: The authors have not demonstrated that exhumation rates are correlated to surface uplift rates or absolute elevation. Line 244, 349: Delete "s" on "uplifts" Line 247: Delete "sea". Line 253–254: This sentence (particularly the second half of the sentence) is unclear. Rewrite. Line 278: Insert "the" before "modern" Line 345: Insert "the" before "late" Line 345: What does "which" refer to? Line 360: This is an awkward sentence. Technically, the middle–late Miocene is part of a geological epoch, not "a fundamental change in earth's (sic) climate system". Line 365: Awkward. Rewrite. Line 397: This is a case in a general point that the authors need to provide more details for readers to be able to evaluate the claims. No mention is made of the depositional setting where this change in CO3 content was observed. Without these details, the reader is forced to find and read the relevant literature, obviating the need for this review. Line 400: What time period are the authors referring to? Hough et al. (2014) observe a regional increase in aridity at ∼14 Ma, but a basin-specific increase in aridity at ∼ 10 Ma. Line 537–542: The timing of the these climatic shifts in the early Cenozoic is not clearly correlative to paleoelevation of the Tibetan Plateau. The southern Tibetan Plateau was probably close to modern elevation in the Late Cretaceous. Why did the climate shift not start then? The only uplift/exhumation events that the authors identify in this timeframe are in the northern Tibetan Plateau. Line 552–554: Again, if these regions were elevated by the Late Cretaceous, what is the driver for initiation of the monsoon in the early Cenozoic? Line 565: Seems ad hoc. CO2 draw-down is attributed to uplift-induced silicate weathering, but upticks in CO2 concentration are unrelated to formation of the Tibetan Plateau? Line 581: The Tibetan Plateau is a big place. What parts of the Tibetan Plateau in specific are the authors talking about?

---

## Author Comment (AC1) · 29 Sep 2018

1. Scientific significance: The authors do not provide new data, analysis, or concepts. The links between silicate weathering and CO2 drawdown have been with us for multiple decades. The discussion of links between tectonic events on the Tibetan Plateau and climate events cherry-picks events and muddles their timing. In this study, we only summarized that the evidence of the age of the Tibetan Plateau, and the climatic variations of inland Asia. On this basis, we found that there are three distinct uplifts in the Tibetan Plateau. At the same time, the climate in Asia had a significant drying trend. Therefore, we proposed that the three main uplifts of the Tibetan Plateau

significantly affect the Asian inland aridification. Therefore, this manuscript was not provided new data and analysis. 2. Major comments : The authors need to clearly delineate time periods of interest. For example, currently all middle–late Miocene climate changes are lumped together, even though the original authors discriminate different mechanisms for climate changes within this period. Similarly, the Cretaceous paleoelevation history of the plateau is largely ignored. See comment on line 400 for an example. Documented uplift events and paleoelevation are not clearly correlative to climate shifts. For example see comment on lines 537–542. Data and conclusions are apparently reported largely without context or comment. This may be fine for an annotated bibliography, but for a review paper, some context and analysis of claims is needed. Otherwise, readers might as well go read all of the citations for themselves. The lack of critical analysis of the data obviates the need for this review. See comments on lines 68 or 397 for an example. Alternative explanations for the observed climate change are dismissed out of hand, without presentation of counter-evidence. We agree with the Reviewer suggestions. We have lumped together middle-late Miocene climate changes because numerous geological evidences show that the Tibetan Plateau has significant outward growth and uplifts of marginal mountains. In the original manuscript, we have emphasized the Cretaceous paleoelevation history (line 189-199). In the revised manuscript, we do not judge whether the results of the published papers are the right or wrong. We only summarized and concluded the evidence in the uplifts of the Tibetan Plateau and associated with climatic changes. Therefore, we do not comment, analyze and declare data and conclusions from the referenced articles. 3. Delete "the" before "Tibet" Line 16, 28, 517: Delete "the" before "uplift" Line 19, 26, 77, 277, 343: Replace "during" with "from" Line 20: Delete "n" in "Himalayan" Line 23: Delete "the" before "Eocene" Line 23–24: Delete "the" before "northern" Line 25: Delete "the" before "central" Line 39: The Cenozoic is an era. Done 4. Line 57: Why "was interpreted"? Reconsider verb tense. Corrected. 5. Line 66: Add "the" before "Lhasa" Line 68: What about the detrital zircon geochronology suggests that Indo- Asian collision occurred at that time? The authors need to provide sufficient

detail for readers to evaluate the claims without having to read the cited literature in its entirety. Corrected. The detrital zircon geochronology was used to constrain firmly the time when Asian-derived detritus was first deposited onto India. We have revised in the manuscript. 6. Line 69–72: This sentence does not make sense. Rewrite. We have deleted. 7. Line 80: Replace "activities" with "activity" Done 8. Line 82: Awkward. Rewrite. Corrected 9. Line 83: No caps on "Paleomagnetic" Corrected 10. Line 86: What paleomagnetic results? See comment on line 68. We have interpreted in the manuscript. The paleomagnetic results are from 43 sites of late Cretaceous red beds, 32 sites of late Cretaceous lava flows, and nine sites of Eocene tuffs. 11.Line 94: Begin this sentence with, "The second is: : :." Done 12. Line 111: What "simulated" data? Line 115: Why "continued"? Is the plateau still being uplifted? Shortening simulation data and corrected it. We have deleted the "continued". 13.Line 121–124: This is not accurate. In the models of Boos and Kuang, the Himalaya do not act as a heat pump, but rather act as a barrier which prevents cold and dry Asian air masses from penetrating southward into the Indian subcontinent. We have not said act as a heat pump and suggested orographic insulation. We have added a word "respectively" in the revised manuscript. 14. Line 139: This whole line of argumentation ignores recent evidence for high elevations in southern Tibet by the Late Cretaceous (e.g., Ding et al., 2014). We have emphasized and suggested in the original manuscript (line 189-199), and not ignored these results. 15. Line 143: Replace "by providing" with "due to" Done 16. Line 152: References? Lines Corrected. 17. 157–158: Basins and the Monsoon cannot have fossil leaf trait spectra. Rewrite. This is the result from the article (Spicer et al., 2016, EPSL) 18. Line 180: Delete "second" Done 19. Line 189: DeCelles et al. (2002) is not an appropriate reference here. Deleted. 20. Line 195: van Hinsbergen et al. (2011) do not present any balanced crosssections. Deleted. 21. Line 230: Change "offer a large amount for" to "would provide abundant". Done 22. Line 236: Delete "the" before "thermal" Done 23. Line 237: The authors have not demonstrated that exhumation rates are correlated to surface uplift rates or absolute elevation. Corrected. 24. Line 244, 349: Delete "s" on "uplifts" Line 247: Delete "sea".

Line 253–254: This sentence (particularly the second half of the sentence) is unclear. Rewrite. Done 25. Line 278: Insert "the" before "modern" Done 26. Line 345: Insert "the" before "late" Done 27. Line 345: What does "which" refer to? "which" refers to the retreat of the Paratethys Sea. 28. Line 360: This is an awkward sentence. Technically, the middle–late Miocene is part of a geological epoch, not "a fundamental change in earth's (sic) climate system". Corrected 29. Line 365: Awkward. Rewrite. Corrected 30. Line 397: This is a case in a general point that the authors need to provide more details for readers to be able to evaluate the claims. No mention is made of the depositional setting where this change in CO3 content was observed. Without these details, the reader is forced to find and read the relevant literature, obviating the need for this review. Corrected. 31. Line 400: What time period are the authors referring to? Hough et al. (2014) observe a regional increase in aridity at $14Ma$, $but a basin-specific increase in aridity at 10Ma$. $We suggested the age of 14-8Ma interval$. $We have emphasized the manuscript$. $32$. $Line. 537--542$ : $The timing of the these climatic shifts in the early Cenozoic is not clearly correlative to paleoelevation of the Tibetan Plateau$. $The scale South Asian summer monsoon circulation is unaffected (Boos and Kuang, 2010, Nature)$. $Therefore, we guess that the so$ $-554$ : $Again, if these regions were elevated by the Late Cretaceous, what is the driver for initiation of the monsoon in the early C$ $Seems ad hoc$. $CO2 draw-down is attributed to uplift-induced silicate weathering, but upticks in CO2 concentration are unrelat$ $The Tibetan Plateau is a big place$. $What parts of the Tibetan Plateau in specific are the authors talking about?$ $The northern marg$

$Please also note the supplement to this comment$ :
$https: //www.clim-past-discuss.net/cp-2018-64/cp-2018-64-AC1-supplement.pdf$

**Supplement:**

Reviewer#1

1. Scientific significance: The authors do not provide new data, analysis, or concepts. The links between silicate weathering and CO2 drawdown have been with us for multiple decades. The discussion of links between tectonic events on the Tibetan Plateau and climate events cherry-picks events and muddles their timing.

In this study, we only summarized that the evidence of the age of the Tibetan Plateau, and the climatic variations of inland Asia. On this basis, we found that there are three distinct uplifts in the Tibetan Plateau. At the same time, the climate in Asia had a significant drying trend. Therefore, we proposed that the three main uplifts of the Tibetan Plateau significantly affect the Asian inland aridification. Therefore, this manuscript was not provided new data and analysis.

2. Major comments : The authors need to clearly delineate time periods of interest. For example, currently all middle–late Miocene climate changes are lumped together, even though the original authors discriminate different mechanisms for climate changes within this period. Similarly, the Cretaceous paleoelevation history of the plateau is largely ignored. See comment on line 400 for an example. Documented uplift events and paleoelevation are not clearly correlative to climate shifts. For example see comment on lines 537–542. Data and conclusions are apparently reported largely without context or comment. This may be fine for an annotated bibliography, but for a review paper, some context and analysis of claims is needed. Otherwise, readers might as well go read all of the citations for themselves. The lack of critical analysis of the data obviates the need for this review. See comments on lines 68 or 397 for an example. Alternative explanations for the observed climate change are dismissed out of hand, without presentation of counter-evidence.

We agree with the Reviewer suggestions. We have lumped together middle-late Miocene climate changes because numerous geological evidences show that the Tibetan Plateau has significant outward growth and uplifts of marginal mountains. In the original manuscript, we have emphasized the Cretaceous paleoelevation history (line 189-199). In the revised manuscript, we do not judge whether the results of the published papers are the right or wrong. We only summarized and concluded the evidence in the uplifts of the Tibetan Plateau and associated with climatic changes. Therefore, we do not comment, analyze and declare data

and conclusions from the referenced articles.

3. Delete "the" before "Tibet" Line 16, 28, 517: Delete "the" before "uplift" Line 19, 26, 77, 277, 343: Replace "during" with "from" Line 20: Delete "n" in "Himalayan" Line 23: Delete "the" before "Eocene" Line 23–24: Delete "the" before "northern" Line 25: Delete "the" before "central" Line 39: The Cenozoic is an era.

Done

4. Line 57: Why "was interpreted"? Reconsider verb tense.

Corrected.

5. Line 66: Add "the" before "Lhasa" Line 68: What about the detrital zircon geochronology suggests that Indo- Asian collision occurred at that time? The authors need to provide sufficient detail for readers to evaluate the claims without having to read the cited literature in its entirety.

Corrected. The detrital zircon geochronology was used to constrain firmly the time when Asian-derived detritus was first deposited onto India. We have revised in the manuscript.

6. Line 69–72: This sentence does not make sense. Rewrite.

We have deleted.

7. Line 80: Replace "activities" with "activity"

Done

8. Line 82: Awkward. Rewrite.

Corrected

9. Line 83: No caps on "Paleomagnetic"

Corrected

10. Line 86: What paleomagnetic results? See comment on line 68.

We have interpreted in the manuscript. The paleomagnetic results are from 43 sites of late Cretaceous red beds, 32 sites of late Cretaceous lava flows, and nine sites of Eocene tuffs.

11.Line 94: Begin this sentence with, "The second is: : :."

Done

12. Line 111: What "simulated" data?   Line 115: Why "continued"? Is the plateau still being uplifted?

Shortening simulation data and corrected it. We have deleted the "continued".

13.Line 121–124: This is not accurate. In the models of Boos and Kuang, the Himalaya do not act as a heat pump, but rather act as a barrier which prevents cold and dry Asian air masses from penetrating southward into the Indian subcontinent.

We have not said act as a heat pump and suggested orographic insulation. We have added a word "respectively" in the revised manuscript.

14. Line 139: This whole line of argumentation ignores recent evidence for high elevations in southern Tibet by the Late Cretaceous (e.g., Ding et al., 2014).

We have emphasized and suggested in the original manuscript (line 189-199), and not ignored these results.

15. Line 143: Replace "by providing" with "due to"

Done

16. Line 152: References? Lines

Corrected.

17. 157–158: Basins and the Monsoon cannot have fossil leaf trait spectra. Rewrite.

This is the result from the article (Spicer et al., 2016, EPSL)

18. Line 180: Delete "second"

Done

19. Line 189: DeCelles et al. (2002) is not an appropriate reference here.

Deleted.

20. Line 195: van Hinsbergen et al. (2011) do not present any balanced crosssections.

Deleted.

21. Line 230: Change "offer a large amount for" to "would provide abundant".

Done

22. Line 236: Delete "the" before "thermal"

Done

23. Line 237: The authors have not demonstrated that exhumation rates are correlated to surface uplift rates or absolute elevation.

Corrected.

24. Line 244, 349: Delete "s" on "uplifts" Line 247: Delete "sea". Line 253–254: This sentence (particularly the second half of the sentence) is unclear. Rewrite.

Done

25. Line 278: Insert "the" before "modern"

Done

26. Line 345: Insert "the" before "late"

Done

27. Line 345: What does "which" refer to?

"which" refers to the retreat of the Paratethys Sea.

28. Line 360: This is an awkward sentence. Technically, the middle–late Miocene is part of a geological epoch, not "a fundamental change in earth's (sic) climate system".

Corrected

29. Line 365: Awkward. Rewrite.

Corrected

30. Line 397: This is a case in a general point that the authors need to provide more details for readers to be able to evaluate the claims. No mention is made of the depositional setting where this change in CO3 content was observed. Without these details, the reader is forced to find and read the relevant literature, obviating the need for this review.

Corrected.

31. Line 400: What time period are the authors referring to? Hough et al. (2014) observe a regional increase in aridity at _14 Ma, but a basin-specific increase in aridity at _ 10 Ma.

We suggested the age of 14-8 Ma interval. We have emphasized the manuscript.

32. Line. 537–542: The timing of the these climatic shifts in the early Cenozoic is not clearly correlative to paleoelevation of the Tibetan Plateau. The southern Tibetan Plateau was probably close to modern elevation in the Late Cretaceous. Why did the climate shift not start then? The only uplift/exhumation events that the authors identify in this timeframe are in the northernTibetan Plateau.

We agree with the opinion that the southern Tibetan Plateau obtained elevation close to modern, whereas the climate is relatively humid in the late Cretaceous. This is an interested problem. Recent models show that mountain uplifts in the northern margin of the Tibetan Plateau have caused significant reductions in annual precipitation in a broad region of inland Asia (Liu et al., 2015, QSR). Also, simulation results show that if we removed of the Tibetan

Plateau, and only provided a narrow orography of the Himalayas and adjacent mountain ranges, the large-scale South Asian summer monsoon circulation is unaffected (Boos and Kuang, 2010, Nature). Therefore, we guess that the southern Tibet uplift has a limited effect on climatic changes in inland Asia. However, this is only guess, and needs further simulations.

33. Line 552–554: Again, if these regions were elevated by the Late Cretaceous, what is the driver for initiation of the monsoon in the early Cenozoic?

Simulation results show that initiation of the monsoon in the early Cenozoic was produced by insulating warm, moist air over continental India from the cold and dry extratropics via the high Himalayas and adjacent mountain ranges (Boos and Kuang, 2010, Nature), but this is need to further simulate in future.

34. Line 565: Seems ad hoc. $CO_2$ draw-down is attributed to uplift-induced silicate weathering, but upticks in $CO_2$ concentration are unrelated to formation of the Tibetan Plateau?

YES. We not suggest that upticks in $CO_2$ content are related to Tibetan Plateau. We say that the upticks in $CO_2$ content during late Oligocene are not related to the silicate weathering.

35. Line 581: The Tibetan Plateau is a big place. What parts of the Tibetan Plateau in specific are the authors talking about?

The northern margins of the Tibetan Plateau. We have emphasized it in the manuscript.

---

## Author Comment (AC2) · 30 Sep 2018

1. Line 428 is an example of a common theme in this work in which the authors relay data from a previous study and state what the data imply, but lack additional background necessary for the reader to understand. In this specific example, the authors cite a "positive shift" in oxygen isotopic values and say that these shifts "imply an increased regional aridification and related to enhanced East Asian winter monsoon." However, no where in the work do they explain how oxygen isotopes are related to aridity or how they can be used to make inferences about atmospheric circulation and weather patterns.

We did not have a detailed discussion how their results were obtained for each of our referenced articles. If we did detailed discussion, this review would be very lengthy. Therefore, our review only summarizes common results, and obtains a basic understanding through their current research. We found that the margins of the Tibetan Plateau have three main uplifts and outward-growth, coeval with the climatic drying in Asian inland. Therefore, we concluded that the main uplifts of the margins of the Tibetan play an important role in climatic changes in inland Asia. In the revised manuscript, we reinterpreted that how oxygen isotopes are related to aridity.

2. Incorrect use of jargon with respect to stable isotopes: I cannot speak to the discussion of paleomagnetism and radiogenic isotope techniques in this work, but I would caution some of the language used with respect to stable isotopes. The authors say "more positive/negative" or "positive/negative shifts" multiple times. A value is positive or negative and cannot be more or less positive or negative. A molecule can have a lighter/heavier isotopic composition w.r.t. a specific isotope/element, or have a lower or higher value. This may seem like a small thing, but will unecessarily irk some readers.

Thanks for reviewer suggestions, and we revised them in the manuscript.

3. Synthesis: The discussion section mostly summarized everything outlined in Section 2 without much additional discussion of the data or contradicting studies. I think for this work to be useful for the community, it should include a more substantive addition to the discourse rather than just a fairly comprehensive laundry list of recent results. Further, the last paragraph of the Discussion calls into question what has come in light of the authors' study of the recent literature. Lines 594-596 state that the authors could not draw linkages between the uplift of the TP and evolution of Asia's climate, which seems to be the motivation of the entire study. If this is true, what has been learned? In the same paragraph, the authors say that climate models do not take into account "detailed topography", but in addition to other such climate modeling work, the authors cite multiple studies that use topographic boundary conditions to constrain the effect of TP uplift on global and regional climate (as recent as the previous paragraph even). I

think the Discussion section would be much improved if this paragraph was removed and replaced with a synthesis of the use of different types of proxies in each of the three tectonic intervals: which proxies seem to agree between the intervals? Which work best and which have greater uncertainty? If the authors believe more studies are needed on topographic boundary conditions, during which intervals and in which sedimentary basins and/or orogenies? These types of questions and answers can help guide the community, which is the ultimate goal of a review paper.

Thanks for reviewer that provided so many suggestions in the Discussion. In Discussion, we have summarized the factors affecting the drought in Aisa, and concluded that the Tibetan Plateau play an important role in Asian aridification during these three intervals, especially during ∼55-35 Ma. We are not going to talk about whose outcome is clearly at odds with the other outcomes. We are just coming up with a basic understanding based on a review of the recent results. But, the contradictions of recent studies are worth studying. We cannot distinguish effects on Asian aridification between global cooling and the Tibetan Plateau during 15-8 Ma interval because of significant global cooling during this period. Because there is still a vague understanding of the uplift height of the Tibetan Plateau, especially the marginal mountains of the Tibetan, therefore, some models are based only on assumptions with respect to altitude in Tibet, which may not be consistent with reality. As a result, the results of the simulation may be uncertain. The reviewer provided a grand goal to decipher the uplift effect and evaluate the best and uncertainty of recent results. This may be beyond the scope and subject of this study. But it is worth exploring in future.

4. Figures Figures 1, 2, and 4: It is useful to see geographically and from what tectonic domains the data used in your interpretations of "rejuvenation or initiation of tectonic activities" comes from, but because different proxies were used in each of the studies marked on the maps, and each proxy records a different thermal regime/extension/rotation/magnetism/etc., it's unclear to me how the different points on the map can be related by the viewer. This ties back into my overall comment

that the reader needs more background on the commonly used techniques in many of your cited studies to assess what each proxy actually records under the umbrella of "rejuvenation of tectonic activities".

We have provided the tables in the manuscript. The tables provided detailed events, ages, methods and references. Therefore, the reader can be obtained the proxy actually records via table 1, 2 and 3.

5. Figure 8: It would be interesting to see some of the additional proxies plotted in Figures 3, 5, and 6 throughout the span of the Cenozoic along with the benthic foraminifera oxygen isotopic composition. The oxygen curve in Figure 8 has been replicated and discussed in numerous studies since its original publication by Zachos, so it would be more intriguing to see how the other proxies change or do not change during the three pulses you attribute climate change and aridification to. Do pedogenic carbonate d18O and wt. % CaCO3 also follow the benthic foram ïAËŻd'18O curve?

We chose additional oxygen isotopes at each interval instead of publication by Zachos (Fig 8) in Figure 3, 5, and 6 because these can provide higher resolution data. The pedogenic carbonate $\delta$18O and wt. %CaCO3 in Tibetan Basins are not coeval with the changes of the benthic foraminiferal $\delta$18O curve. There is a lag time about ~2 Myr. The significant decrease of benthic foraminiferal $\delta$18O curve occurred at ~13.9 Ma (Figure 5), but the significant increase or decrease of the pedogenic carbonate $\delta$18O and wt. %CaCO3, respectively, occurred at about ~12 Ma. This difference may indicate that another factor, such as tectonic uplift of marginal mountains in Tibet, plays a role in climatic changes.

Please also note the supplement to this comment:
https://www.clim-past-discuss.net/cp-2018-64/cp-2018-64-AC2-supplement.pdf